# THE MECHANISTIC EMERGENCE OF SYMBOL GROUNDING IN LANGUAGE MODELS

## ABSTRACT

Symbol grounding (Harnad, 1990) describes how symbols such as words acquire their meanings by connecting to real-world sensorimotor experiences. Recent work has shown preliminary evidence that grounding may emerge in (vision-)language models trained at scale without using explicit grounding objectives. Yet, the specific loci of this emergence and the mechanisms that drive it remain largely unexplored. To address this problem, we introduce a controlled evaluation framework that systematically traces how symbol grounding arises within the internal computations through mechanistic and causal analysis. Our findings show that grounding concentrates in middle-layer computations and is implemented through the aggregate mechanism, where attention heads aggregate the environmental ground to support the prediction of linguistic forms. This phenomenon replicates in multimodal dialogue and across architectures (Transformers and state-space models), but not in unidirectional LSTMs. Our results provide behavioral and mechanistic evidence that symbol grounding can emerge in language models, with practical implications for predicting and potentially controlling the reliability of generation.

## 1 INTRODUCTION

Symbol grounding (Harnad, 1990) refers to the problem of how abstract and discrete symbols, such as words, acquire meaning by connecting to perceptual or sensorimotor experiences. Extending to the context of multimodal machine learning, grounding has been leveraged as an explicit pre-training objective for vision-language models (VLMs), by explicitly connecting linguistic units to the world that gives language meanings (Li et al., 2022; Ma et al., 2023). Through supervised fine-tuning with grounding signals, such as entity-phrase mappings, modern VLMs have achieved fine-grained understanding at both region (You et al., 2024; Peng et al., 2024; Wang et al., 2024) and pixel (Zhang et al., 2024b; Rasheed et al., 2024; Zhang et al., 2024a) levels.

With the rising of powerful autoregressive language models (LMs; OpenAI, 2024; Anthropic, 2024; Comanici et al., 2025, *inter alia*) and their VLM extensions, there is growing interest in identifying and interpreting their emergent capabilities. Recent work has shown preliminary correlational evidence that grounding may emerge in LLMs (Sabet et al., 2020; Shi et al., 2021; Wu et al., 2025) and VLMs (Cao et al., 2024; Bousselham et al., 2024; Schnaus et al., 2025) trained at scale, even when solely optimized with the simple next-token prediction objective. However, the potential underlying mechanisms that lead to such an emergence are not well understood. To address this limitation, our work seeks to understand the emergence of symbol grounding in LMs, causally and mechanistically tracing how symbol grounding arises within the internal computations.

We begin by constructing a minimal testbed, motivated by the annotations provided in the CHILDES corpora (MacWhinney, 2000), where child–caregiver interactions provide cognitively plausible contexts for studying symbol grounding alongside verbal utterances. In our framework, each word is represented in two distinct forms: one token that appears in non-verbal scene descriptions (e.g., a *box* in the environment) and another that appears in spoken utterances (e.g., *box* in dialogue). We refer to these as environmental tokens ($\langle$ENV$\rangle$) and linguistic tokens ($\langle$LAN$\rangle$), respectively. A deliberately simple word-level tokenizer assigns separate vocabulary entries to each form, ensuring that they are treated as entirely different tokens by the language model. This framework enforces a structural separation between scenes and symbols, preventing correspondences from being reduced to trivial token identity. Under this setup, we can evaluate whether a model trained from scratch is able to predict the linguistic form from its environmental counterpart.

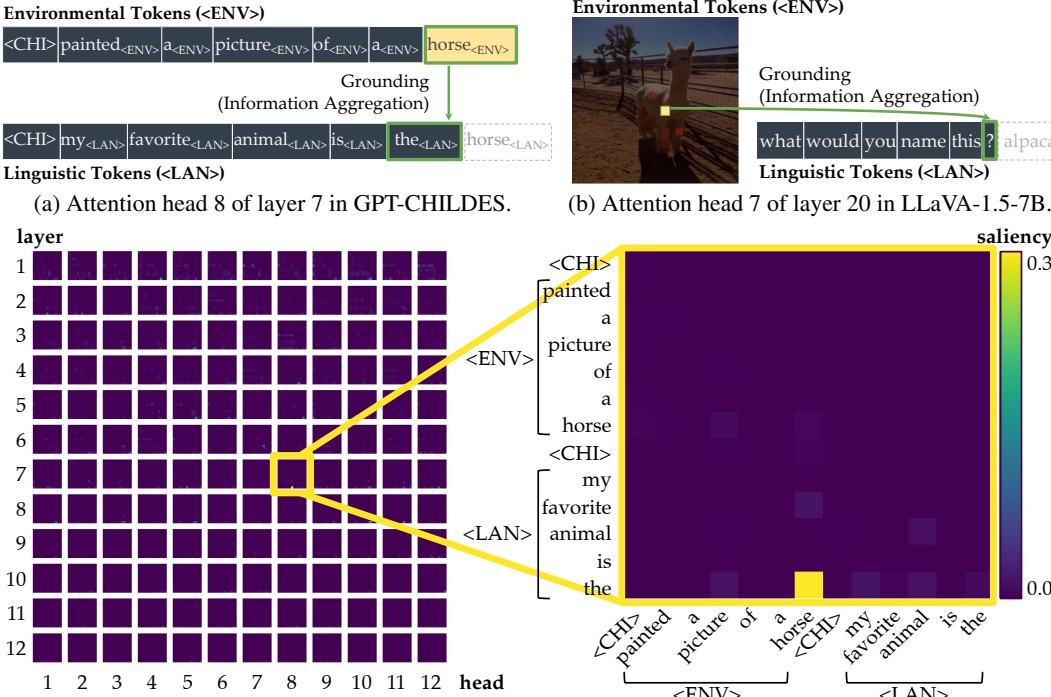

(a) Attention head 8 of layer 7 in GPT-CHILDES.    (b) Attention head 7 of layer 20 in LLaVA-1.5-7B.

(c) Left: saliency over tokens of each head in each layer for the prompt $\langle CHI \rangle$ $painted_{\langle ENV \rangle}$ $a_{\langle ENV \rangle}$ $picture_{\langle ENV \rangle}$ $of_{\langle ENV \rangle}$ $a_{\langle ENV \rangle}$ $horse_{\langle ENV \rangle}$ $\langle CHI \rangle$ $my_{\langle LAN \rangle}$ $favorite_{\langle LAN \rangle}$ $animal_{\langle LAN \rangle}$ $is_{\langle LAN \rangle}$ $the_{\langle LAN \rangle}$. Right: among all, only one of them (head 8 of layer 7) is identified as an aggregate head, where information flows from $horse_{\langle ENV \rangle}$ to the current position, encouraging the model to predict $horse_{\langle LAN \rangle}$ as the next token.

Figure 1: Illustration of the symbol grounding mechanism through information aggregation. Lighter colors denote more salient attention, quantified by saliency scores, i.e., gradient × attention contributions to the loss (Wang et al., 2023). When predicting the next token, aggregate heads (Bick et al., 2025) emerge to exclusively link environmental tokens (visual or situational context; $\langle ENV \rangle$) to linguistic tokens (words in text; $\langle LAN \rangle$). These heads provide a mechanistic pathway for symbol grounding by mapping external environmental evidence into its linguistic form.

We quantify the level of grounding using surprisal: specifically, we compare how easily the model predicts a linguistic token ($\langle LAN \rangle$) when its matching environmental token ($\langle ENV \rangle$) is present versus when unrelated cues are given instead. A lower surprisal in the former condition indicates that the model has learned to align environmental grounds with linguistic forms. We find that LMs do learn to ground: the presence of environmental tokens consistently reduces surprisal for their linguistic counterparts, in a way that simple co-occurrence statistics cannot fully explain. To study the underlying mechanisms, we apply saliency analysis (Wang et al., 2023) and the tuned lens (Belrose et al., 2023), which converge on the result that grounding relations are concentrated in the middle layers of the network. Further analysis of attention heads reveals patterns consistent with the aggregate mechanism (Bick et al., 2025), where attention heads support the prediction of linguistic forms by retrieving their environmental grounds in the context.

Finally, we demonstrate that these findings generalize beyond the minimal CHILDES data and Transformer models. They appear in a multimodal setting with the Visual Dialog dataset (Das et al., 2017), and in state-space models (SSMs) such as Mamba-2 (Dao & Gu, 2024). In contrast, we do not observe grounding in unidirectional LSTMs, consistently with their sequential state compression and lack of content-addressable retrieval. Taken together, our results show that symbol grounding can mechanistically emerge in autoregressive LMs, while also delineating the architectural conditions under which it can arise.

## 2    RELATED WORK

**Language grounding.** Referential grounding has long been framed as the lexicon acquisition problem: how words map to referents in the world (Harnad, 1990; Gleitman & Landau, 1994;

Clark, 1995). Early work focused on word-to-symbol mappings, designing learning mechanisms that simulate children's lexical acquisition and explain psycholinguistic phenomena (Siskind, 1996; Regier, 2005; Goodman et al., 2007; Fazly et al., 2010). Subsequent studies incorporated visual grounding, first by aligning words with object categories (Roy & Pentland, 2002; Yu, 2005; Xu & Tenenbaum, 2007; Yu & Ballard, 2007; Yu & Siskind, 2013), and later by mapping words to richer visual features (Qu & Chai, 2010; Mao et al., 2019; 2021; Pratt et al., 2020). More recently, large-scale VLMs trained with paired text–image supervision have advanced grounding to finer levels of granularity, achieving region-level (Li et al., 2022; Ma et al., 2023; Chen et al., 2023; You et al., 2024; Wang et al., 2024) and pixel-level (Xia et al., 2024; Rasheed et al., 2024; Zhang et al., 2024b) grounding, with strong performance on referring expression comprehension (Chen et al., 2024a).

Recent work suggests that grounding emerges as a property of VLMs trained without explicit supervision, with evidence drawn from attention-based spatial localization (Cao et al., 2024; Bousselham et al., 2024) and cross-modal geometric correspondences (Schnaus et al., 2025). However, all prior work focused exclusively on static final-stage models, overlooking the training trajectory, a crucial aspect for understanding when and how grounding emerges. In addition, existing work has framed grounding through correlations between visual and textual signals, diverging from the definition by Harnad (1990), which emphasizes causal links from symbols to meanings. To address these issues, we systematically examine learning dynamics throughout the training process, applying causal interventions to probe model internals and introducing control groups to enable rigorous comparison.

**Emergent capabilities and learning dynamics of LMs.** A central debate concerns whether larger language models exhibit genuinely new behaviors: Wei et al. (2022) highlight abrupt improvements in tasks, whereas later studies argue such effects are artifacts of thresholds or in-context learning dynamics (Schaeffer et al., 2023; Lu et al., 2023). Beyond end performance, developmental analyses show that models acquire linguistic abilities in systematic though heterogeneous orders with variability across runs and checkpoints (Sellam et al., 2021; Blevins et al., 2022; Biderman et al., 2023; Xia et al., 2023; van der Wal et al., 2025). Psychology-inspired perspectives further emphasize controlled experimentation to assess these behaviors (Hagendorff, 2023), and comparative studies reveal both parallels and divergences between machine and human language learning (Chang & Bergen, 2022; Evanson et al., 2023; Chang et al., 2024; Ma et al., 2025). At a finer granularity, hidden-loss analyses identify phase-like transitions (Kangaslahti et al., 2025), while distributional studies attribute emergence to stochastic differences across training seeds (Zhao et al., 2025). Together, emergent abilities are not sharp discontinuities but probabilistic outcomes of developmental learning dynamics. Following this line of work, we present a probability- and model internals–based analysis of how symbol grounding emerges during language model training.

**Mechanistic interpretability of LMs.** Mechanistic interpretability has largely focused on attention heads in Transformers (Elhage et al., 2021; Olsson et al., 2022; Meng et al., 2023; Bietti et al., 2023; Lieberum et al., 2023; Wu et al., 2024). A central line of work established that *induction heads* emerge to support in-context learning (ICL; Elhage et al., 2021; Olsson et al., 2022), with follow-up studies tracing their training dynamics (Bietti et al., 2023) and mapping factual recall circuits (Meng et al., 2023). At larger scales, Lieberum et al. (2023) identified specialized *content-gatherer* and *correct-letter* heads, and Wu et al. (2024) showed that a sparse set of *retrieval heads* is critical for reasoning and long-context performance. Relatedly, Wang et al. (2023) demonstrated that label words in demonstrations act as *anchors*: early layers gather semantic information into these tokens, which later guide prediction. Based on these insights, Bick et al. (2025) proposed that retrieval is implemented through a coordinated *gather-and-aggregate (G&A)* mechanism: some heads collect content from relevant tokens, while others aggregate it at the prediction position. Other studies extended this line of work by analyzing failure modes and training dynamics (Wiegreffe et al., 2025) and contrasting retrieval mechanisms in Transformers and SSMs (Arora et al., 2025). Whereas prior analyses typically investigate ICL with repeated syntactic or symbolic formats, our setup requires referential alignment between linguistic forms and their environmental contexts, providing a complementary testbed for naturalistic language grounding.

## 3 METHOD

### 3.1 DATASET AND TOKENIZATION

To capture the emergent grounding from multimodal interactions, we design a minimal testbed with a custom word-level tokenizer, in which every lexical item is represented in two corresponding forms:

one token that appears in non-verbal descriptions (e.g., a *book* in the scene description) and another that appears in utterances (e.g., *book* in speech). We refer to these by environmental ($\langle\text{ENV}\rangle$) and linguistic tokens ($\langle\text{LAN}\rangle$), respectively. For instance, $book_{\langle\text{ENV}\rangle}$ and $book_{\langle\text{LAN}\rangle}$ are treated as distinct tokens with separate integer indices; that is, the tokenization provides no explicit signal that these tokens are related, so any correspondence between them must be learned during training rather than inherited from their surface form. We instantiate this framework in three datasets, ranging from child-directed speech transcripts to image-based dialogue.

**Child-directed speech.** The Child Language Data Exchange System (CHILDES; MacWhinney, 2000) provides transcripts enriched with environmental annotations.[1] We use the spoken utterances as the linguistic tokens ($\langle\text{LAN}\rangle$) and the environmental descriptions as the environment tokens ($\langle\text{ENV}\rangle$). The environmental context is drawn from three annotation types:

- **Local events**: simple events, pauses, long events, or remarks interleaved with the transcripts.
- **Action tiers**: actions performed by the speaker or listener (e.g., `%act: runs to toy box`). These also include cases where an action replaces speech (e.g., `0 [% kicks the ball]`).
- **Situational tiers**: situational information tied to utterances or to larger contexts (e.g., `%sit: dog is barking`).

**Caption-grounded dialogue.** The Visual Dialog dataset (Das et al., 2017) pairs MSCOCO images (Lin et al., 2014) with sequential question-answering based multi-turn dialogues that exchange information about each image. Our setup uses MSCOCO captions as the environmental tokens ($\langle\text{ENV}\rangle$) and the dialogue turns form the linguistic tokens ($\langle\text{LAN}\rangle$). In this pseudo cross-modal setting, textual descriptions of visual scenes ground natural conversational interaction. Compared to CHILDES, this setup introduces richer semantics and longer utterances, while still using text-based inputs for both token types, thereby offering a stepping stone toward grounding in fully visual contexts.

**Image-grounded dialogue.** To move beyond textual proxies, we consider an image-grounded dialogue setup, using the same dataset as the caption-grounded dialogue setting. Here, a frozen vision transformer (ViT; Dosovitskiy et al., 2020) directly tokenizes each RGB image into patch embeddings, with each embedding treated as an $\langle\text{ENV}\rangle$ token, analogously to the visual tokens in modern VLMs. We use DINOv2 (Oquab et al., 2024) as our ViT tokenizer, as it is trained purely on vision data without auxiliary text supervision (in contrast to models like CLIP; Radford et al., 2021), thereby ensuring that environmental tokens capture only visual information. The linguistic tokens ($\langle\text{LAN}\rangle$) remain unchanged from the caption-grounded dialogue setting, resulting in a realistic multimodal interaction where conversational utterances are grounded directly in visual input.

## 3.2 EVALUATION PROTOCOL

We assess symbol grounding with a contrastive test that asks whether a model assigns a higher probability to the correct linguistic token when the matching environmental token is in context, following the idea of priming in psychology. This evaluation applies uniformly across datasets (Table 1): in CHILDES and caption-grounded dialogue, environmental priming comes from descriptive contexts; in image-grounded dialogue, from ViT-derived visual tokens. We compare the following conditions:

- **Match (experimental condition)**: The context contains the corresponding $\langle\text{ENV}\rangle$ token for the target word, and the model is expected to predict its $\langle\text{LAN}\rangle$ counterpart.
- **Mismatch (control condition)**: The context is replaced with a different $\langle\text{ENV}\rangle$ token. The model remains tasked with predicting the same $\langle\text{LAN}\rangle$ token; however, in the absence of corresponding environmental cues, its performance is expected to be no better than chance.

For example (first row in Table 1), when evaluating the word $book_{\langle\text{LAN}\rangle}$, the input context is

$$\langle CHI \rangle \; asked_{\langle\text{ENV}\rangle} \; for_{\langle\text{ENV}\rangle} \; a_{\langle\text{ENV}\rangle} \; new_{\langle\text{ENV}\rangle} \; book_{\langle\text{ENV}\rangle} \; \langle CHI \rangle \; I_{\langle\text{LAN}\rangle} \; love_{\langle\text{LAN}\rangle} \; this_{\langle\text{LAN}\rangle} \; \underline{\quad\quad}, \quad (1)$$

where the model is expected to predict $book_{\langle\text{LAN}\rangle}$ for the blank, and the role token $\langle CHI \rangle$ indicates the involved speaker or actor's role being a child. In the control (mismatch) condition, the environmental token $box_{\langle\text{ENV}\rangle}$ is replaced by another valid noun such as $toy_{\langle\text{ENV}\rangle}$.

**Context templates.** For a target word $v$ with linguistic token $v_{\langle\text{LAN}\rangle}$ and environmental token $v_{\langle\text{ENV}\rangle}$, we denote $\overline{C}_v$ as a set of context templates of $v$. For example, when $v = book$, a $\overline{c} \in \overline{C}_v$ can be

$$\underline{\quad\quad} \langle CHI \rangle \; asked_{\langle\text{ENV}\rangle} \; for_{\langle\text{ENV}\rangle} \; a_{\langle\text{ENV}\rangle} \; new_{\langle\text{ENV}\rangle} \; [\text{FILLER}] \; \langle CHI \rangle \; I_{\langle\text{LAN}\rangle} \; love_{\langle\text{LAN}\rangle} \underline{\quad\quad}, \quad (2)$$

---

[1] See the manual for data usage: `https://talkbank.org/0info/manuals/CHAT.pdf`

Table 1: Training and test examples across datasets with target word *book*. The training examples combine environmental tokens ( $\langle$ENV$\rangle$; shaded ) with linguistic tokens ($\langle$LAN$\rangle$). Test examples are constructed with either matched (*book*) or mismatched (*toy*) environmental contexts, paired with corresponding linguistic prompts. Note that in child-directed speech and caption-grounded dialogue, $book_{\langle ENV \rangle}$ and $book_{\langle LAN \rangle}$ are two distinct tokens received by LMs.

| Dataset | Training Example | | Test Example | | |
| --- | --- | --- | --- | --- | --- |
| | $\langle$ENV$\rangle$ | $\langle$LAN$\rangle$ | $\langle$ENV$\rangle$ Match | $\langle$ENV$\rangle$ Mismatch | $\langle$LAN$\rangle$ |
| Child-Directed Speech | $\langle CHI \rangle$ *takes book from mother* | $\langle CHI \rangle$ *what's that* $\langle MOT \rangle$ *a book in it ...* | $\langle CHI \rangle$ *asked for a new book* | $\langle CHI \rangle$ *asked for a new toy* | $\langle CHI \rangle$ *I love this* ———— |
| Caption-Grounded Dialogue | *a dog appears to be reading a book with a full bookshelf behind* | $\langle Q \rangle$ *can you tell what book it's reading* $\langle A \rangle$ *the marriage of true minds by stephen evans* | *this is a book* | *this is a toy* | $\langle Q \rangle$ *can you name this object* $\langle A \rangle$ ———— |
| Image-Grounded Dialogue |  | $\langle Q \rangle$ *can you tell what book it's reading* $\langle A \rangle$ *the marriage of true minds by stephen evans* |  |  | *what do we have here?* ———— |

where [FILLER] is to be replaced with an environmental token, and the blank indicates the expected prediction as in Eq. (1). In the match condition, the context $\bar{c}(v)$ is constructed by replacing [FILLER] with $v_{\langle ENV \rangle}$ in $\bar{c}$. In the mismatch condition, the context $\bar{c}(u)$ uses $u_{\langle ENV \rangle}(u \neq v)$ as the filler, while the prediction target remains $v_{\langle LAN \rangle}$.

For the choices of $v$ and $u$, we construct the vocabulary $V$ with 100 nouns from the MacArthur–Bates Communicative Development Inventories (Fenson et al., 2006) that occur frequently in our corpus. Each word serves once as the target, with the remaining $M = 99$ used to construct mismatched conditions. For each word, we create $N = 10$ context templates, which contain both $\langle$ENV$\rangle$ and $\langle$LAN$\rangle$ tokens. Details of the vocabulary and context template construction can be found in the Appendix A.

**Grounding information gain.** Following prior work, we evaluate how well an LM learns a word using the mean surprisal over instances. The surprisal of a word $w$ given a context $c$ is defined as $s_{\boldsymbol{\theta}}(w \mid c) = -\log P_{\boldsymbol{\theta}}(w \mid c)$, where $P_{\boldsymbol{\theta}}(w \mid c)$ denotes the probability, under an LM parameterized by $\boldsymbol{\theta}$, that the next word is $w$ conditioned on the context $c$. Here, $s_{\boldsymbol{\theta}}(w \mid c)$ quantifies the unexpectedness of predicting $w$, or the pointwise information carried by $w$ conditioned on the context $c$.

The *grounding information gain* $G_{\boldsymbol{\theta}}(v)$ for $v$ is defined as

$$G_{\boldsymbol{\theta}}(v) = \frac{1}{N} \sum_{n=1}^{N} \left( \frac{1}{M} \sum_{u \neq v}^{M} \left[ s_{\boldsymbol{\theta}} \left( v_{\langle LAN \rangle} \mid \bar{c}_n \left( u_{\langle ENV \rangle} \right) \right) - s_{\boldsymbol{\theta}} \left( v_{\langle LAN \rangle} \mid \bar{c}_n \left( v_{\langle ENV \rangle} \right) \right) \right] \right).$$

This is a sample-based estimation of the expected log-likelihood ratio between the match and mismatch conditions

$$G_{\boldsymbol{\theta}}(v) = \mathbb{E}_{c,u} \left[ \log \frac{P_{\boldsymbol{\theta}}(v_{\langle LAN \rangle} \mid c, v_{\langle ENV \rangle})}{P_{\boldsymbol{\theta}}(v_{\langle LAN \rangle} \mid c, u_{\langle ENV \rangle})} \right],$$

which quantifies how much more information the matched ground provides for predicting the linguistic form, compared to a mismatched one. A positive $G_{\boldsymbol{\theta}}(v)$ indicates that the matched environmental token increases the predictability of its linguistic form. We report $G_{\boldsymbol{\theta}} = \frac{1}{|V|} \sum_{v \in V} G_{\boldsymbol{\theta}}(v)$, and track $G_{\boldsymbol{\theta}^{(t)}}$ across training steps $t$ to analyze how grounding emerges over time.

## 3.3 MODEL TRAINING

We train LMs from random initialization, ensuring that no prior linguistic knowledge influences the results. Our training uses the standard causal language modeling objective, as in most generative LMs. To account for variability, we repeat all experiments with 5 random seeds, randomizing both model initialization and corpus shuffle order. Our primary architecture is Transformer (Vaswani et al., 2017) in the style of GPT-2 (Radford et al., 2019) with 18, 12, and 4 layers, with all of them having residual

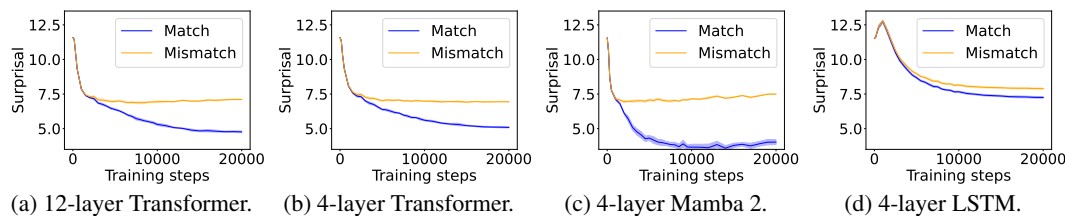

(a) 12-layer Transformer.    (b) 4-layer Transformer.    (c) 4-layer Mamba 2.    (d) 4-layer LSTM.

Figure 2: Average surprisal of the experimental and control conditions over training steps.

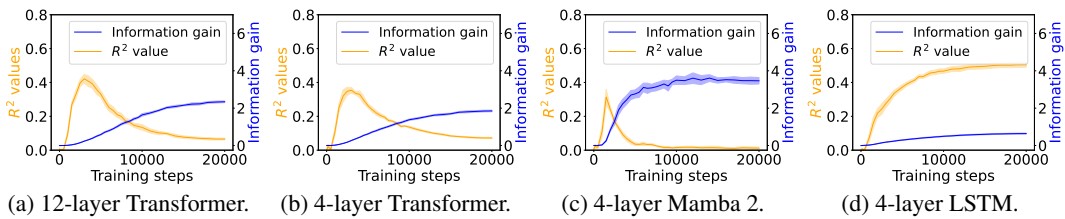

(a) 12-layer Transformer.    (b) 4-layer Transformer.    (c) 4-layer Mamba 2.    (d) 4-layer LSTM.

Figure 3: Grounding information gain and its correlation to the co-occurrence of linguistic and environment tokens over training steps.

connections. We extend the experiments to 4-layer unidirectional LSTMs (Hochreiter & Schmidhuber, 1997) with no residual connections, as well as 12- and 4-layer state-space models (specifically, Mamba-2; Dao & Gu, 2024). For fair comparison with LSTMs, the 4-layer Mamba-2 models do not involve residual connections, whereas the 12-layer ones do. For multimodal settings, while standard LLaVA (Liu et al., 2023) uses a two-layer perceptron to project ViT embeddings into the language model, we bypass this projection in our case and directly feed the DINOv2 representations into the LM. We obtain the developmental trajectory of the model by saving checkpoints at various training steps, sampling more heavily from earlier steps, following Chang & Bergen (2022).

## 4 BEHAVIORAL EVIDENCE

### 4.1 BEHAVIORAL EVIDENCE OF EMERGENT GROUNDING

In this section, we ask: **Does symbol grounding emerge behaviorally in autoregressive LMs?** We first test whether models show systematic surprisal reduction when predicting a linguistic token when its environmental counterpart is in context (Figure 2, where the gap between the lines represent the grounding information gain). For Transformers (Figures 2a and 2b) and Mamba-2 (Figure 2c), surprisal in the match condition decreases steadily while that in the mismatch condition enters a high-surprisal plateau early, indicating that the models leverage environmental context to predict the linguistic form. In contrast, the unidirectional LSTM (Figure 2d) shows little separation between the conditions, reflecting the absence of grounding. Overall, these results provide behavioral evidence of emergent grounding: in sufficiently expressive architectures (Transformers and Mamba-2), the correct environmental context reliably lowers surprisal for its linguistic counterpart, whereas LSTMs fail to exhibit this effect, marking an architectural boundary on where grounding can emerge.

### 4.2 BEHAVIORAL EFFECTS BEYOND CO-OCCURRENCE

A natural concern is that the surprisal reductions might be fully explainable by shallow statistics: **the models might have simply memorized frequent co-occurrences of ⟨ENV⟩ and ⟨LAN⟩ tokens, without learning a deeper and more general mapping.** We test this hypothesis by comparing the tokens' co-occurrence with the grounding information gain in the child-directed speech data.

We define co-occurrence between the corresponding ⟨ENV⟩ and ⟨LAN⟩ tokens at the granularity of a 512-token training chunk. For each target word $v$, we count the number of chunks in which both its ⟨ENV⟩ and ⟨LAN⟩ tokens appear. Following standard corpus-analysis practice, these raw counts are log-transformed. For each model checkpoint, we run linear regression between the log co-occurrence and the grounding information gain of words, obtaining an $R^2$ statistic as a function of training time.

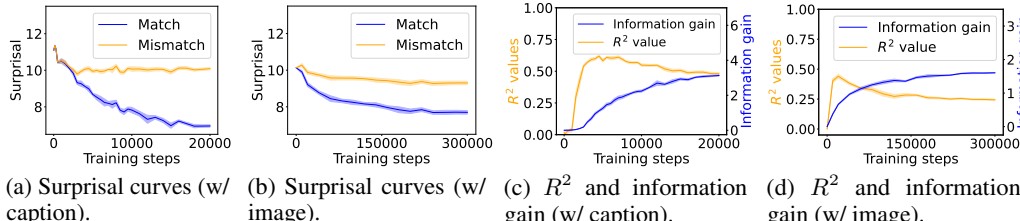

(a) Surprisal curves (w/ caption).  (b) Surprisal curves (w/ image).  (c) $R^2$ and information gain (w/ caption).  (d) $R^2$ and information gain (w/ image).

Figure 4: Average surprisal of the experimental and control conditions, as well as the grounding information gain and its correlation to the co-occurrence of linguistic and environment tokens over training steps. All results are from a 12-layer Transformer model on grounded dialogue data.

Figure 3 shows the $R^2$ values (orange) alongside the grounding information gain (blue) for different architectures. In both the Transformer and Mamba-2, $R^2$ rises sharply at the early steps but then goes down, even if the grounding information gain continues increasing. These results suggest that grounding in Transformers and Mamba-2 cannot be fully accounted for by co-occurrence statistics: while models initially exploit surface co-occurrence regularities, later improvements in grounding diverge from these statistics, indicating reliance on richer and more complicated features acquired during training. In contrast, LSTM shows persistently increasing $R^2$ but little increase in grounding information gain over training steps, suggesting that it encodes co-occurrence but lacks the architectural mechanism to transform it into predictive grounding.

### 4.3 Visual Dialogue with Captions and Images

We next test whether the grounding effects observed in CHILDES generalize to multimodal dialogue, using the Visual Dialog dataset. In this setting, the environmental ground is supplied either by captions or by image features (Table 1). For caption-grounded dialogue, the mismatch context is constructed in the same way as for CHILDES (Equation 2). For image-grounded dialogue, mismatch contexts are generated via Stable Diffusion 2 (Rombach et al., 2022)–based image inpainting, which re-generates the region defined by the ground-truth mask corresponding to the target word's referent.

We train 12-layer Transformers with 5 random seeds. Similarly as Figures 2a–2b and Figures 3a–3b, when captions serve as the environmental ground, Transformers show a clear surprisal gap between match and mismatch conditions (Figure 4a), with the grounding information gain increasing steadily while $R^2$ peaks early and declines (Figure 4c). Directly using image as grounds yields the same qualitative pattern (Figures 4b and 4d), although the observed effect is smaller. Both settings confirm that emergent grounding cannot be fully explained by co-occurrence statistics.

Overall, our findings demonstrate that Transformers are able to exploit environmental grounds in various modalities to facilitate linguistic prediction. The smaller but consistent gains in the image-grounded case suggest that while grounding from visual tokens is harder, the same architectural dynamics identified in textual testbeds still apply.

## 5 Mechanistic Explanation

In this section, we provide a mechanistic and interpretable account of the previous observation. We first focus on a 12-layer Transformer trained on CHILDES with 5 random seeds, and extend the experiments to image-grounded dialogue (Section 5.4).

### 5.1 The Emergence of Symbol Grounding

To provide a mechanistic account of symbol grounding, i.e., when it emerges during training and how it is represented in the network, we apply two interpretability analyses.

**Saliency flow.** For each layer $\ell$, we compute a saliency matrix following Wang et al. (2023): $I_\ell = \left| \sum_h A_{h,\ell} \odot \frac{\partial \mathcal{L}}{\partial A_{h,\ell}} \right|$, where $A_{h,\ell}$ denotes the attention matrix of head $h$ in layer $\ell$. Each entry of $I_\ell$ quantifies the contribution of the corresponding attention weight to the cross-entropy loss $\mathcal{L}$, averaged across heads. Our analysis focuses on ground-to-symbol connections, i.e., flows from environmental ground ($\langle \mathsf{ENV} \rangle$) tokens to the token immediately preceding (and predicting) their linguistic forms ($\langle \mathsf{LAN} \rangle$).

**Probing with the Tuned Lens.** We probe layer-wise representations using the Tuned Lens (Belrose et al., 2023), which trains affine projectors to map intermediate activations to the final prediction space while keeping the LM output head frozen.

**Results.** Ground-to-symbol saliency is weak in the early stages of training but rises sharply later, peaking in layers 7–9 (Figure 5a), suggesting that mid-layer attention plays a central role in establishing symbol–ground correspondences. In addition, Figure 5b shows that early layers remain poor predictors even at late training stages (e.g., after 20,000 steps), whereas surprisal begins to drop markedly from layer 7 at intermediate stages (step 10,000), suggesting a potential representational shift in the middle layers.

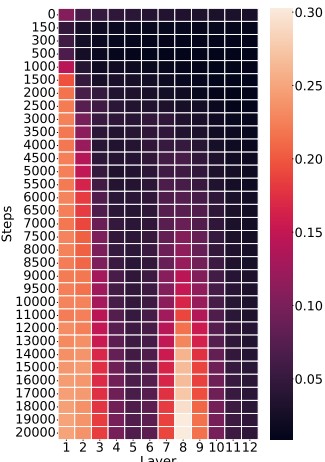

(a) Saliency of layer-wise attention from environmental to linguistic tokens across training steps.

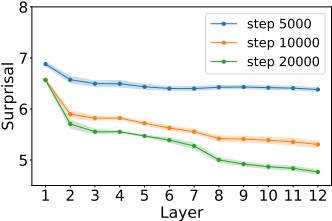

(b) Layer-wise tuned lens to predict the ⟨LAN⟩ token in match condition.

Figure 5: Mechanistic analysis of symbol grounding emergence.

## 5.2 HYPOTHESIS: GATHER-AND-AGGREGATE HEADS IMPLEMENT SYMBOL GROUNDING

Building on these results, we hypothesize that specific Transformer heads in the middle layers enable symbol grounding. To test this, we examine attention saliencies for selected heads (Figure 6). We find that several heads exhibit patterns consistent with the gather and aggregate mechanisms described by Bick et al. (2025): gather heads (e.g., Figures 6a and 6b) compress relevant information into a subset of positions, while aggregate heads (e.g., Figures 6c and 6d) redistribute this information to downstream tokens. In our setups, saliency often concentrates on environmental tokens such as $train_{\langle ENV \rangle}$, where gather heads pool contextual information into compact, retrievable states. In turn, aggregate heads broadcast this information from environmental ground ($train\langle ENV \rangle$) to the token immediately preceding the linguistic form, thereby supporting the prediction of $train_{\langle LAN \rangle}$. Taking these observations together, we hypothesize that the gather-and-aggregate heads implement the symbol grounding mechanism.

## 5.3 CAUSAL INTERVENTIONS OF ATTENTION HEADS

We then conduct causal interventions of attention heads to validate our previous hypothesis.

**Operational definition.** We identify attention heads as gather or aggregate following these standards:

- **Gather head.** An attention head is classified as a gather head if at least 30% of its total saliency is directed toward the environmental ground token from the previous ones.

- **Aggregate head**: An attention head is classified as an aggregate head if at least 30% of its total saliency flows from the environmental ground token to the token immediately preceding the corresponding linguistic token.

**Causal intervention methods.** In each context, we apply causal interventions to the identified head types and their corresponding controls. Following Bick et al. (2025), interventions are implemented by zeroing out the outputs of heads. For the control, we mask an equal number of randomly selected heads in each layer, ensuring they do not overlap with the identified gather or aggregate heads.

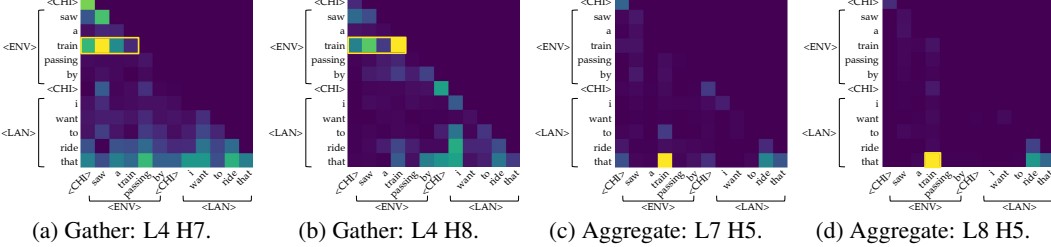

(a) Gather: L4 H7.  (b) Gather: L4 H8.  (c) Aggregate: L7 H5.  (d) Aggregate: L8 H5.

Figure 6: Examples of gather and aggregate heads identified. L: layer; H: head.

**Results and discussions.** As training progresses, the number of both gather and aggregate heads increases (Table 2), suggesting that these mechanisms emerge over the course of learning. Causal interventions reveal a clear dissociation: zeroing out aggregate heads consistently produces significantly higher surprisal compared to controls, whereas the gather head interventions have no such effect. This asymmetry suggests that gather heads serve in a role less critical in our settings, where the input template is semantically light and the environmental evidence alone suffices to shape the linguistic form. Layer-wise patterns further support this division of labor: gather heads cluster in shallow layers (3-4), while aggregate heads concentrate in mid layers (7-8). This resonates with our earlier probing results, where surprisal reductions became prominent only from layers 7-9. Together, these findings highlight aggregate heads in the middle layers as the primary account of grounding in the model.

## 5.4 GENERALIZATION TO VISUAL DIALOG WITH IMAGES

We also conduct causal interventions of attention heads on the VLM model to further validate our previous hypothesis.

**Operational definition.** We identify attention heads as aggregate following this standard (We do not define gather head): An attention head is classified as an aggregate head if at least a certain threshold (70% or 90% in our experiment settings) of its total image patch to end saliency flows from the patches inside bounding box to the token immediately preceding the corresponding linguistic token.

**Causal intervention methods.** In each context, we apply causal interventions to the identified head types and their corresponding controls in the language backbone of the model. Similar to section 5.3, interventions are implemented by zeroing out a head's outputs. For the control, we mask an equal number of randomly selected heads in each layer, ensuring they do not overlap with the identified aggregate heads.

Table 2: Causal intervention results on identified gather and aggregate heads across training checkpoints (ckpt.). **Avg. Count** denotes the average number of heads of each type over inference times, and **Avg. Layer** denotes the average layer index where they appear. **Interv. Sps.** reports surprisal after zeroing out the identified heads, while **Ctrl. Sps.** reports surprisal after zeroing out an equal number of randomly selected heads. **Original** refers to the baseline surprisal without any intervention. *** indicates a significant result ($p < 0.001$) where the intervention surprisal is higher than that in the corresponding control experiment.

| Ckpt. | Gather Head | | | | Aggregate Head | | | | Original |
|---|---|---|---|---|---|---|---|---|---|
| | Avg. Count | Avg. Layer | Interv. Sps. | Ctrl. Sps. | Avg. Count | Avg. Layer | Interv. Sps. | Ctrl. Sps. | |
| 500 | 0.00 | - | - | - | 0.07 | 8.74 | 9.34 | 9.34 | 9.34 |
| 5000 | 0.35 | 3.32 | 6.37 | 6.38 | 2.28 | 7.38 | **6.51** (***) | 6.39 | 6.38 |
| 10000 | 3.26 | 3.67 | 5.25 | 5.32 | 5.09 | 7.28 | **5.86** (***) | 5.29 | 5.30 |
| 20000 | 5.76 | 3.59 | 4.69 | 4.79 | 6.71 | 7.52 | **5.62** (***) | 4.76 | 4.77 |

| Thres. | Ckpt. | Aggregate Head | | | | Original |
|---|---|---|---|---|---|---|
| | | Avg. Count | Avg. Layer | Interv. Sps. | Ctrl. Sps. | |
| 70% | 20k | 32.30 | 7.78 | 9.96 | 9.95 | 9.21 |
| | 100k | 35.63 | 7.71 | **9.42** (***) | 8.84 | 8.24 |
| | 200k | 34.99 | 7.80 | **8.95** (***) | 8.15 | 7.76 |
| | 300k | 34.15 | 7.76 | **8.96** (***) | 8.11 | 7.69 |
| 90% | 20k | 10.66 | 8.33 | **9.51** (***) | 9.43 | 9.21 |
| | 100k | 13.90 | 8.26 | **8.95** (***) | 8.50 | 8.24 |
| | 200k | 13.47 | 8.46 | **8.41** (***) | 7.88 | 7.76 |
| | 300k | 12.73 | 8.42 | **8.40** (***) | 7.87 | 7.69 |

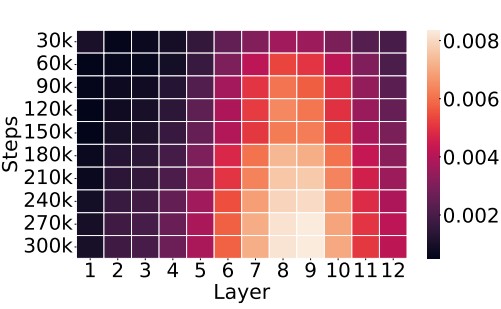

Figure 7: Mechanistic analysis in the image-grounded visual dialogue setting. Left: Causal intervention results on identified aggregate heads across training checkpoints, where intervention on aggregate heads consistently yields significantly higher surprisal ($p < 0.001$, ***) compared to the control group ones. Right: Saliency of layer-wise attention from environmental tokens (i.e., image tokens corresponding to patches within the bounding boxes of the target object) to linguistic tokens across training steps.

**Results and discussions.** As training progresses, the number of aggregate heads increases first and then becomes steady (Figure 7), suggesting that these mechanisms emerge over the course of learning. Causal interventions reveal that zeroing out aggregate heads consistently produces significantly higher surprisal rises compared to controls. The average layer also align with the saliency heatmap, also shown in Figure 7.

## 6 DISCUSSIONS

**Generalization to full-scale VLMs.** As an additional case study, we extend our grounding-as-aggregation hypothesis to a full-scale VLM, LLaVA-1.5-7B (Liu et al., 2023). Even in this heavily engineered architecture, we identify many attention heads exhibiting aggregation behavior consistent with our earlier findings (Figure 1b), reinforcing the view that symbol grounding arises from specialized heads. At the same time, full-scale VLMs present additional complications. Models like LLaVA use multiple sets of visual tokens, including CLIP-derived embeddings that already encode language priors, and global information may be stored in redundant artifact tokens rather than object-centric regions (Darcet et al., 2024). Moreover, the large number of visual tokens (environmental tokens, in our setup) substantially increases both computational cost and the difficulty of isolating genuine aggregation heads. These factors make systematic identification and intervention at scale a nontrivial challenge. For these reasons, while our case study highlights promising evidence of grounding heads in modern VLMs, systematic detection and causal evaluation of such heads at scale remains an open challenge. Future work will need to develop computationally viable methods for (i) automatically detecting aggregation heads across diverse VLMs, and (ii) applying causal interventions to validate their role in grounding.

**Connection to symbol binding.** While our primary purpose of introducing the minimal testbed (child-direct speech setting; Table 1) is to offer a proxy for understanding grounding, experiments in this setting can be naturally viewed as investigations of symbol binding, the problem that studies how symbols are connected together. This work extends the activation-based study on symbol binding (Feng & Steinhardt, 2024; Dai et al., 2024; Feng et al., 2025) and offers evidence on the attention-head level, showing that aggregate heads are crucial in implementing the mechanism (Table 2). In line with Yang et al. (2025), our work suggests that attention heads are crucial in implementing symbolic structures in LMs, with more controlled and causal evidence.

**The philosophical roots of grounding, revisited.** Our findings highlight the need to sharpen the meaning of grounding in multimodal models. Prior work has often equated grounding with statistical correlations between visual and textual signals, such as attention overlaps or geometric alignments (Cao et al., 2024; Bousselham et al., 2024; Schnaus et al., 2025). While informative, such correlations diverge from the classic formulation by Harnad (1990), which requires symbols to be causally anchored to their referents in the environment. In line with Harnad (1990), we frame grounding as a mechanistic property: one that can be traced along training, observed in the specialization of attention heads, and validated through causal interventions, providing a protocol for diagnosing when and how models genuinely tie symbols to meaning rather than mere correlations. On another line, our results, which show that aggregate heads implement symbol grounding (and binding), echo Pavlick (2023) in arguing that *LLMs lack the capacity to represent abstract symbolic structure* should not be accepted a priori. Instead, such claims should be evaluated carefully and empirically, with the focus on uncovering the models' underlying competence rather than drawing conclusions solely from high-level architectures and surface-level performance.

**Practical implications to LM hallucinations.** Our findings have practical implications for improving the reliability of LM outputs: by identifying aggregation heads that mediate grounding between environmental and linguistic tokens, we provide a promising mechanism to detect model reliability before generation. Our findings echo a pathway to mitigate hallucinations by focusing on attention control: many hallucination errors stem from misallocated attention in intermediate layers (Jiang et al., 2025; Chen et al., 2024b). Such attention-level signals can serve as early indicators of overtrust or false grounding, motivating practical solutions like decoding-time strategies to mitigate and eventually prevent hallucination (Huang et al., 2024).

ETHICS STATEMENT

This study analyzes the mechanistic emergence of symbol grounding in language models using publicly available datasets such as CHILDES and Visual Dialog. All the images are from the publicly available MSCOCO dataset (Lin et al., 2014) where no personally identifiable data is used.

REPRODUCIBILITY STATEMENT

We describe dataset construction, tokenization schemes, evaluation protocols, and model training settings in detail within the main text and Appendix. All experiments were repeated with multiple random seeds, and results across different architectures are reported. Code and data processing scripts are included in supplementary materials and will be released upon acceptance to facilitate full reproducibility.

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

# A   DATASET DETAILS

## A.1   CONTEXT TEMPLATES

We select the target tokens following the given procedure:

1. Get a list of words, with their ENV and LAN frequency both greater than or equal to 100 in the CHILDES dataset;
2. Get another list of nouns from CDI;
3. Take intersection and select top 100 words (by frequency of their ENV token) as target token list.

In CHILDES, all contexts are created with `gpt-4o-mini` followed by human verification if the genrated contexts are semantically light. We adopt the following prompt:

---

**Prompt Templates for CHILDES**

```
Given the word "{word}", create 3 pairs of sentences that follow this
requirement:
1. The first sentence has a subject "The child", describing an event or
situation, and has the word "{word}". Make sure to add a newline to the end of
this first sentence
2. The second sentence is said by the child (only include the speech itself,
don't include "the child say", etc.), and the word "{word}" also appears in
the sentence said by the child. Do not add quote marks either
3. Print each sentence on one line. Do not include anything else.
4. Each sentence should be short, less than 10 words.
5. The word "{word}" in both sentence have the same meaning and have a clear
indication or an implication relationship.
6. "{word}" should not appear at the first/second word of each sentence.
Generate 3 pairs of such sentences, so there should be 6 lines in total.
You should not add a number.
For each line, just print out the sentence.
```

---

In visual dialogue (caption version and VLM version), we pre-define 10 sets of templates for each version:

---

**Prompt Templates for Visual Dialogue (Caption Version)**

```
this:<ENV> is:<ENV> [FILLER]:<ENV> <Q> what:<LAN> is:<LAN> it:<LAN> <A>
(predict [FILLER]:<LAN>)

this:<ENV> is:<ENV> [FILLER]:<ENV> <Q> what:<LAN> do:<LAN> you:<LAN>
call:<LAN> this:<LAN> <A> (predict [FILLER]:<LAN>)

this:<ENV> is:<ENV> [FILLER]:<ENV> <Q> can:<LAN> you:<LAN>
name:<LAN> this:<LAN> object:<LAN> <A>
(predict [FILLER]:<LAN>)

this:<ENV> is:<ENV> [FILLER]:<ENV> <Q> what's:<LAN>
this:<LAN> called:<LAN> <A>
(predict [FILLER]:<LAN>)

this:<ENV> is:<ENV> [FILLER]:<ENV> <Q> what:<LAN>
this:<LAN> thing:<LAN> is:<LAN> <A>
(predict [FILLER]:<LAN>)
```

---

**Prompt Templates for Visual Dialogue (Caption Version) (continued)**

```
this:<ENV> is:<ENV> [FILLER]:<ENV> <Q> what:<LAN>
would:<LAN> you:<LAN> name:<LAN> this:<LAN> <A>
(predict [FILLER]:<LAN>)

this:<ENV> is:<ENV> [FILLER]:<ENV> <Q>
what's:<LAN> the:<LAN> name:<LAN> of:<LAN> this:<LAN>
item:<LAN> <A> (predict [FILLER]:<LAN>)

this:<ENV> is:<ENV> [FILLER]:<ENV> <Q> how:<LAN>
do:<LAN> you:<LAN> identify:<LAN> this:<LAN> <A>
(predict [FILLER]:<LAN>)

this:<ENV> is:<ENV> [FILLER]:<ENV> <Q> what:<LAN>
do:<LAN> we:<LAN> have:<LAN> here:<LAN> <A>
(predict [FILLER]:<LAN>)

this:<ENV> is:<ENV> [FILLER]:<ENV> <Q> how:<LAN>
do:<LAN> you:<LAN> call:<LAN> this:<LAN>
object:<LAN> <A> (predict [FILLER]:<LAN>)
```

**Prompt Templates for Visual Dialogue (VLM Version)**

```
"<image> \nwhat is it ?",
"<image> \nwhat do you call this ?",
"<image> \ncan you name this object ?",
"<image> \nwhat is this called ?",
"<image> \nwhat this thing is ?",
"<image> \nwhat would you name this ?",
"<image> \nwhat is the name of this item ?",
"<image> \nhow do you identify this ?",
"<image> \nwhat do we have here ?",
"<image> \nhow do you call this object ?"
```

## A.2 WORD LIST FOR CHILDES AND VISION DIALOGUE (TEXT ONLY)

[box, book, ball, hand, paper, table, toy, head, car, chair, room, picture, doll, cup, towel, door, mouth, camera, duck, face, truck, bottle, puzzle, bird, tape, finger, bucket, block, stick, elephant, hat, bed, arm, dog, kitchen, spoon, hair, blanket, horse, tray, train, cow, foot, couch, necklace, cookie, plate, telephone, window, brush, ear, pig, purse, hammer, cat, shoulder, garage, button, monkey, pencil, shoe, drawer, leg, bear, milk, egg, bowl, juice, ladder, basket, coffee, bus, food, apple, bench, sheep, airplane, comb, bread, eye, animal, knee, shirt, cracker, glass, light, game, cheese, sofa, giraffe, turtle, stove, clock, star, refrigerator, banana, napkin, bunny, farm, money]

## A.3 WORD LIST FOR VISION DIALOGUE (VLM)

[box, book, table, toy, car, chair, doll, door, camera, duck, truck, bottle, bird, elephant, hat, bed, dog, spoon, horse, train, couch, necklace, cookie, plate, telephone, window, pig, cat, monkey, drawer, bear, milk, egg, bowl, juice, ladder, bus, food, apple, sheep, bread, animal, shirt, cheese, giraffe, clock, refrigerator, accordion, aircraft, alpaca, ambulance, ant, antelope, backpack, bagel, balloon, barrel, bathtub, beard, bee, beer, beetle, bicycle, bidet, billboard, boat, bookcase, boot, boy, broccoli, building, bull, burrito, bust, butterfly, cabbage, cabinetry, cake, camel, canary, candle, candy, cannon,

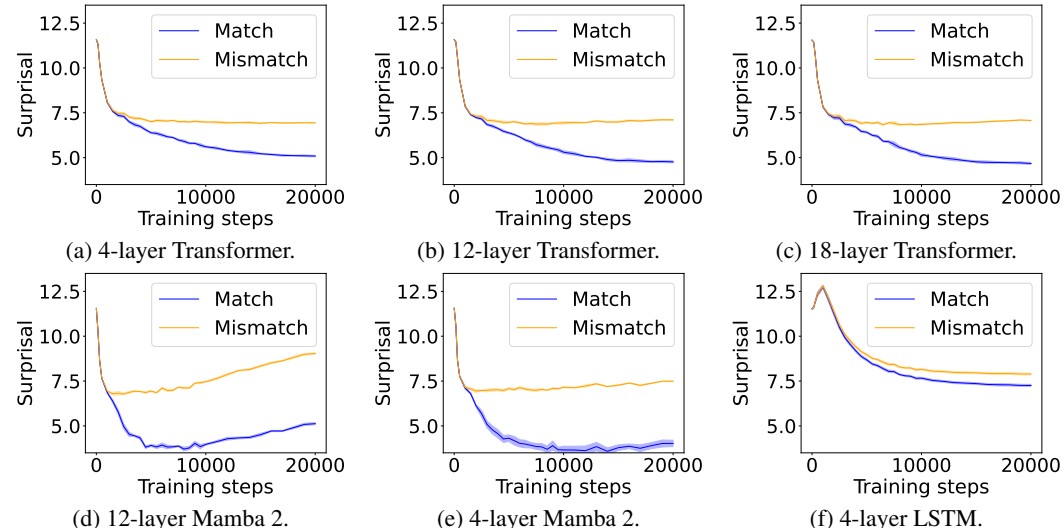

Figure 8: Average surprisal of the experimental and control conditions over training steps.

canoe, carrot, cart, castle, caterpillar, cattle, cello, cheetah, chicken, chopsticks, closet, clothing, coat, cocktail, coffeemaker, coin, cosmetics]

## B  IMPLEMENTATION DETAILS

### B.1  CHECKPOINTING

#### B.1.1  CHILDES, VISUAL DIALOGUE WITH CAPTIONS

We save the intermediate steps: [0, 150, 300, 500, 1000, 1500, 2000, 2500, 3000, 3500, 4000, 4500, 5000, 5500, 6000, 6500, 7000, 7500, 8000, 8500, 9000, 9500, 10000, 11000, 12000, 13000, 14000, 15000, 16000, 17000, 18000, 19000, 20000] (33 checkpoints in total)

#### B.1.2  VISUAL DIALOGUE (VLM)

We save the intermediate steps: [10000, 20000, 40000, 60000, 80000, 100000, 120000, 140000, 160000, 180000, 200000, 220000, 240000, 260000, 280000, 300000] (16 checkpoints in total)

## C  ADDENDUM TO RESULTS

### C.1  DETAILED BEHAVIORAL ANALYSIS FOR ALL MODELS

We show the complete behavioral evidence for all models in Figure 8, and co-occurrence analysis in Figure 9. On top of that, a beeswarm plot indicating per context match/mismatch surprisal is shown in Figure 10.

### C.2  DETAILED GATHER AND AGGREGATE ANALYSIS (TRANSFORMER)

After finding the set of gather and aggregate heads for each context, we run an overtime analysis showing the proportion of saliency to the total saliency, as is shown in Figure 11.

### C.3  RELATIONSHIP BETWEEN AGGREGATE HEAD NUMBER AND GROUNDING INFORMATION GAIN

We examine the 12-layer Transformer LM for both the child-directed speech and VLM settings. For each target token (detailed in Sections A.2 and A.3), we compare its grounding information gain with

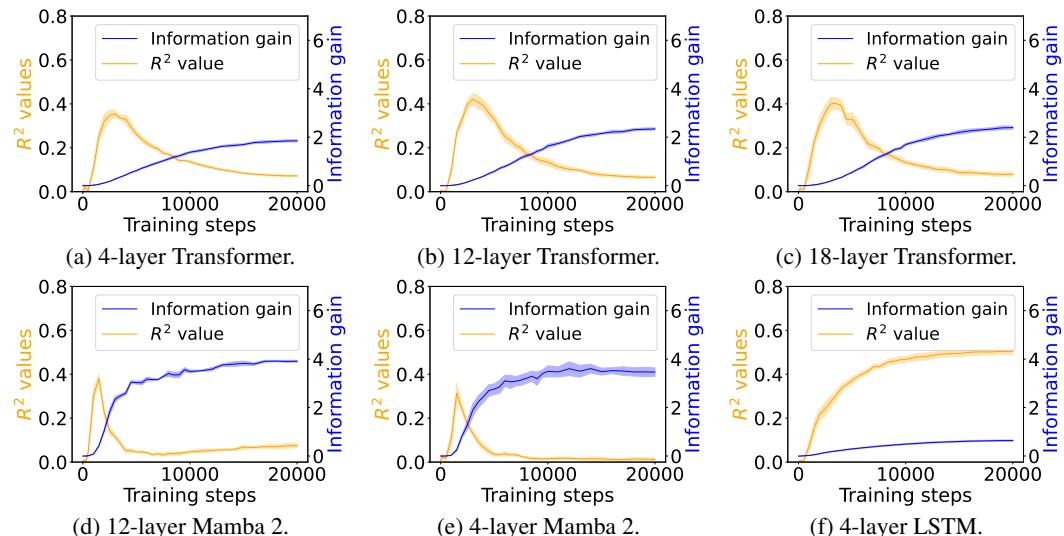

(a) 4-layer Transformer.     (b) 12-layer Transformer.     (c) 18-layer Transformer.

(d) 12-layer Mamba 2.     (e) 4-layer Mamba 2.     (f) 4-layer LSTM.

Figure 9: Grounding information gain and its correlation to the co-occurrence of linguistic and environment tokens over training steps.

the average aggregate head it has (Figure 12). We find an intermediate-to-strong positive correlation between the grounding information gain and the aggregate head number detected.

## D   LLM STATEMENT

In this work, large language models (LLMs) are employed in two limited ways: (i) to polish the writing and improve the linguistic clarity of the paper; (ii) to assist in code writing and debugging. LLMs are not involved in the design of the core method, the experimental setup, data analysis, or the interpretation of the results. All texts presented in the paper, as well as the code, are endorsed by the authors, and the authors take full responsibility of the content presented in this paper.

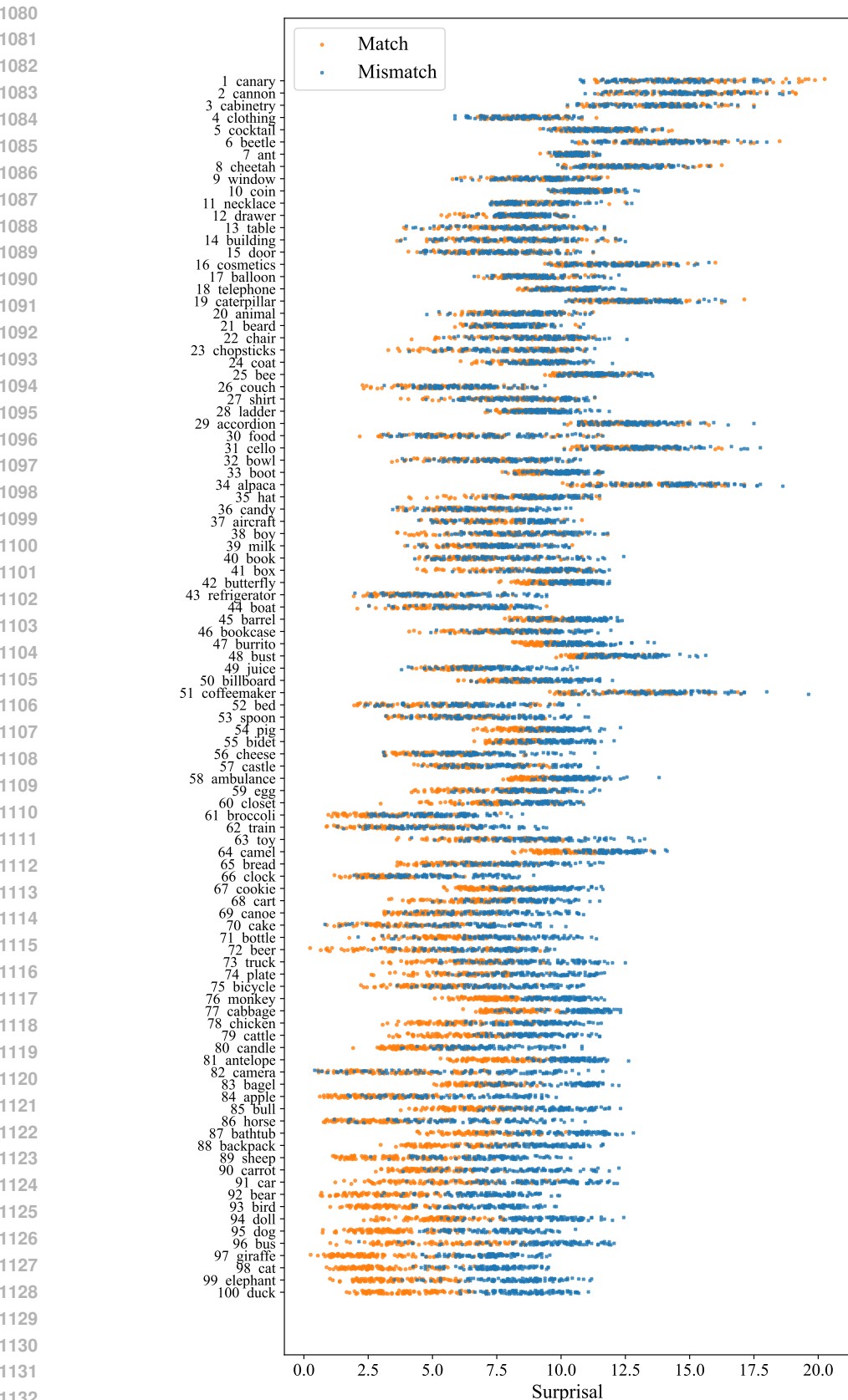

Figure 10: Per-instance surprisal in our trained VLM, sorted by information-gain per word (increasing). Orange dots: surprisals in matched context; blue cross: surprisals in mismatched context.

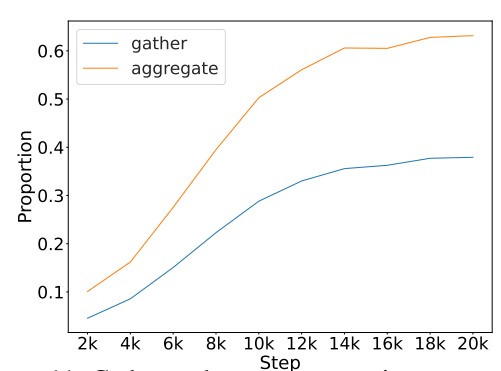

Figure 11: Gather-and-aggregate overtime strength.

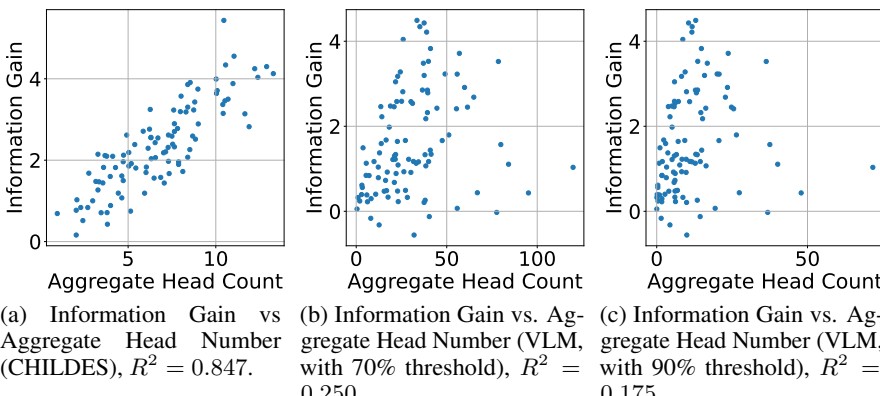

(a) Information Gain vs Aggregate Head Number (CHILDES), $R^2 = 0.847$.

(b) Information Gain vs. Aggregate Head Number (VLM, with 70% threshold), $R^2 = 0.250$.

(c) Information Gain vs. Aggregate Head Number (VLM, with 90% threshold), $R^2 = 0.175$.

Figure 12: Grounding information gain vs. number of aggregate heads.

