# OpenReview forum: "The Mechanistic Emergence of Symbol Grounding in Language Models"
_ICLR.cc/2026/Conference — Submitted to ICLR 2026_

### Official Review · Reviewer_zfMv · 2025-10-25

**Soundness:** 2
**Presentation:** 3
**Contribution:** 2
**Rating:** 2
**Confidence:** 3

**Summary:**

The paper designs tasks where certain context token (environmental token, can be vision token) can be good signal for predicting another token (linguistics token). When trained on these tasks, models can find their connection and make use of it. The authors put this process in the context of symbol grounding in VLMs (how linguistic tokens is grounded to vision patterns). They aim to understand grounding in VLM by studying these simplified and controlled settings. Authors also show clear evidence supporting that such connection differs from normal co-occurrence. They also analyze and show there is a gather and aggregate mechanism in the model that achieve this.

**Strengths:**

- The study uses R-squared statistic to show that the grounding can not be explained by simple co-occurrence.
- The study includes 3 different tasks involving symbol grounding and various model architectures (transformers, SSMs, LSTMs of different layers)

**Weaknesses:**

- The dataset is small, containing only 100 nouns, and templates for constructing text containing \<LAN\> are simple (often without much extra text as noise). This makes the grounding signal potentially much stronger than the case in real world pretraining. Such difference might affect the phenomenon shown in the paper, and can be the key to understand grounding (i.e. how strong does it need to be to make grounding happen)

- The paper uses too much space to show such grounding can emerge in the proposed tasks. But content for investigating how it happens thoughout training (I mean more detailed studies besides just information gain throughout training) and what is the mechanism is limited, while the authors say that the goal is to understand the emergence of grounding.

- The mechanistic interpretation is limited and evidence is vague. The results show some important layers and heads. But the usage of Saliency flow is only an estimation of coarse-grained importance. There’s also categorization of heads according their attention patterns. But they only show higher-than average causal effect. We do see some patterns in Table 2, which shows heads that tend to attend \<ENV\> is more important than average attention head. But this information is limited, it does not mean they or the hypothesized mechanism fully explains the grounding (in that case other heads should have almost zero effects).

**Questions:**

- The paper argues that the main mechanism is gather-and-aggregate, so both the gather heads and aggregate heads should play important roles. But why in table 2, intervening on gather heads does not affect the surprisal? Because they are needed by aggregate heads, zeroing out them should also destroy the circuit.

- The naive co-occurrence cannot explain information gain. Do you think it might be the case that the model just figure out when to use co-occurrence statistics? or which tokens can be predicted from occurrence of some other tokens.  So it just uses it for some tokens, like nouns.

---

> ### Author Response · Authors · 2025-11-21
> **Response to Reviewer zfMv**
>
> We thank the reviewer for their time reviewing our work. We are glad that the reviewer found that “certain context token (environmental token, can be vision token) can be good signal for predicting another token (linguistics token).” We also appreciate the recognition that we provide “clear evidence supporting that such connection differs from normal co-occurrence,” and that our mechanistic analysis identifies and “shows there is a gather and aggregate mechanism in the model that achieve this.”
>
> ### Weakness 1: Training Setup
> We believe there are some factual misunderstandings about our setups. The reviewer seems to have confused the pre-training corpora and test templates evaluated at inference time. We hope we can clarify:
> - **Pretraining data**. The CHILDES dataset (around 22M tokens) and Visual Dialog (around 25M tokens) are used for pretraining a GPT-2 model from scratch. This corpus contains substantial natural noise and exhibits a strong imbalance in both environmental tokens and language tokens. Therefore, it does not “make the grounding signal potentially much stronger than the case in real-world pretraining”, but weaker.
> - **Test templates.** All generated templates are used only for testing. The “100 nouns” refer specifically to the concrete noun subset we selected for evaluation (selection criteria described in Appendix A.1). Importantly, these are not the only nouns appearing in the pretraining corpus. Each noun receives 10 templates, adding to 1000 test instances.
> - **Summary.** In short, our model is pretrained on a realistic, noisy dataset that reflects conditions similar to real-world pretraining. The generated templates serve exclusively as evaluation prompts for measuring grounding ability, and they are never used during pretraining.
>
> ### Weakness 2: "more detailed studies besides just information gain throughout training"
> We respectfully clarify a misunderstanding regarding the allocation of space in the results section. The part the reviewer refers to as “too much space showing that grounding emerges” corresponds only to Section 4.1, which is 11 lines and serves purely as a brief confirmation and problem setup.
>
> The substantial analysis, how grounding emerges and what mechanisms underlie it, is presented in the subsequent sections:
> - §4.2: token co-occurrence vs. grounding information gain (excludes trivial corpus effects)
> - §4.3: generalization to multimodal dialogue (cross-task emergence)
> - §5.1: Tuned Lens trajectories and attention-saliency through training
> - §5.2-5.3: attention-head-level mechanism analysis (which heads encode grounding signals, and how)
>
> These sections together constitute the majority of the paper’s experimental content and directly study the mechanism of emergence throughout training. Therefore, the concern that our paper includes “too much space showing emergence but limited mechanism analysis” appears to be based on a misreading of the structure. We also include several additional experiments and analysis, e.g., on causal intervention and performance v.s. number of effective aggregate heads during the discussion phase.
>
> ### Weakness 3: Limitations of our mechanistic explanation
> The reviewer argues that "the usage of Saliency flow is only an estimation of coarse-grained importance." But section 5 integrates: (i) probing of intermediate representations, (ii) attention-pattern analysis identifying aggregate-head structure, and (iii) ablation studies measuring causal impact. All three independently reveal the same phenomenon: grounding is centrally encoded by aggregate heads in mid layers (L7-8). This is not a single-method observation but a cross-validated mechanism.
>
> The reviewer describes the causal effects as “slightly higher-than-average,” but this is factually incorrect: surprisal is measured on a log scale, and differences such as 5.62 vs. 4.76 correspond to $e^{5.62-4.76}=2.36$× differences in impact, not marginal variations. Thus, the causal significance of aggregate heads is substantially larger than what the raw numbers may appear to suggest.
>
> The reviewer suggests that because other heads do not drop to near-zero effect. We observed that when critical heads are ablated during evaluation, models re-route information and form new aggregate heads in previously non-aggregate heads. We did not include this result because: (i) it expands beyond the scope of this paper, and (ii) it warrants an analysis of its own.

---

> > ### Author Response · Authors · 2025-11-21
> >
> > ### Question 1
> >
> > We first clarify **2 factual misunderstandings** that:
> > - We never "argue that the main mechanism is gather-and-aggregate". Rather, we explicitly present it as an initial hypothesis in Section 5.2, which we then test via causal intervention experiments. These experiments show that aggregate heads have substantially higher causal importance than gather heads.
> > - We do not assume that gather heads are "needed by aggregate heads" in the sense that zeroing gather heads should necessarily destroy the circuit. This is a misinterpretation of Bick et al. (2025), where gather-and-aggregate is proposed as a possible mechanism, not as a claim that both components must always be simultaneously necessary in a strict causal sense.
> >
> > Regarding why aggregate heads are more significant, we provide a detailed discussion in the general response (Question 1.2).
> >
> > Here we offer one possible explanation. We hypothesize that grounding is more of a direct connection process, where information of the ground is directly broadcast to the token immediately preceding the corresponding linguistic token, and aggregate heads naturally implement such a function.
> >
> > Additionally, in our minimal testbed, since the context template is semantically light, the environmental evidence alone may suffice to decide the linguistic form. For example, if the environmental context is `<CHI> holds:<ENV> a:<ENV> book:<ENV>`, token `book:<ENV>` is sufficient to determine the linguistic token (which is very likely to be `book:<LAN>`). In such a setting, gathering information from the context is not necessary, as an aggregate head can perform this "one-hop" retrieval directly.
> >
> > ### Question 2
> > Regarding the reviewer’s suggestion that “the model may simply learn when to use co-occurrence statistics,” we emphasize that this is not an alternative explanation—it is exactly the developmental trajectory we document. Our findings show:
> > - Early in training, information gain aligns with surface-level co-occurrence (a predictable baseline).
> > - Later in training, information gain diverges sharply from any co-occurrence-based predictor (as shown in §4.2), especially for abstract or relational tokens where naive frequency cues fail.
> >
> > This divergence corresponds exactly to the emergence of aggregate heads and mid-layer representational restructuring (§5.1-5.3), indicating a learned mechanism beyond co-occurrence lookup. Thus, the idea that the model “figures out when co-occurrence is useful” is precisely the mechanism of grounding emergence: the model moves from exploiting superficial corpus regularities to forming context-sensitive, structured representations.

---

### Official Review · Reviewer_ZTuF · 2025-10-28

**Soundness:** 3
**Presentation:** 3
**Contribution:** 3
**Rating:** 6
**Confidence:** 4

**Summary:**

This paper claims to causally and mechanistically trace the emergence of symbol grounding in language models. The authors design a two-token system, where each lexical item appears as a linguistic token (\<LAN\>) and an “environmental” token (\<ENV\>), then measure reductions in surprisal when the linked token is present. They further identify “aggregate heads” responsible for propagating information from \<ENV\> to \<LAN\> tokens.

**Strengths:**

Well-structured evaluations and mechanistic interpretability experiments.

Analysis across multiple architectures and modalities is thorough.

Causal ablations offer meaningful insights into attention head specialization.

**Weaknesses:**

The proposed evaluation does not measure symbol grounding:

The setup relies purely on two linguistically implemented tokens that correspond to the same object. The “environmental tokens” are still just symbols with no perceptual grounding.

Thus, the model’s task reduces to:

learning a mapping between symbol A (\<ENV\>) and symbol B (\<LAN\>)

rather than mapping a symbol to real-world referents.
This fails to demonstrate grounding as defined in Harnad (1990), which requires sensorimotor anchoring:

| Aspect | True Symbol Grounding | Paper’s Setup |
|--------|----------------------|---------------|
| Source of Meaning | Perception / world interaction | Another text token |
| Representation Modality | Non-symbolic | Symbolic |
| Link Type | Causal grounding | Token association |


While the reported effects are intriguing, it remains unclear whether the current setup demonstrates symbol grounding per se, as opposed to learning an association between two linked tokens. In particular, the “image-grounded” condition relies on pretrained ViT features that may already encode semantic regularities, which complicates claims of grounding from raw perceptual input. We encourage the authors to provide a more detailed argument for why the two-token design suffices to establish symbol grounding rather than symbol–symbol alignment, and to clarify the causal pathway from environmental evidence to linguistic meaning.

**Questions:**

1. Can the model still “ground” if you remove any shared lexical identity between \<ENV\> and \<LAN\> tokens (e.g., no 1–1 mapping)?

2. Does grounding transfer to unseen objects or contexts? Specifically, information between token and intrinsicly visualized object?

---

> ### Author Response · Authors · 2025-11-21
> **Response to Reviewer ZTuF**
>
> We thank the reviewer for their thoughtful and constructive feedback and are encouraged by their positive assessment of our work. We appreciate that the paper is seen as “causally and mechanistically trace[ing] the emergence of symbol grounding in language models,” and that our evaluations and analyses are characterized as “well-structured evaluations and mechanistic interpretability experiments.” We are also grateful that the “analysis across multiple architectures and modalities is thorough,” and that our “causal ablations offer meaningful insights into attention head specialization.”
>
> ### Weakness 1: The discrepancy between the problem investigated in this work vs. Hanard (1990)
>
> We thank the reviewer for the thoughtful comments. Our explanations and corresponding revisions are provided in the general response (Issue 1). In short, we believe there might be a misunderstanding regarding the purpose of our minimal testbed, the technical implementation, and the non-symbolic nature of "visual tokens" in vision-language models, or both. We have also updated the manuscript with causal intervention results on VLMs.
>
> Particularly, we would like to argue against the comment:
> > In particular, the “image-grounded” condition relies on pretrained ViT features that may already encode semantic regularities, which complicates claims of grounding from raw perceptual input.
>
> We used DINOv2 (Oquab et al., 2024), a vision-only self-supervised model, as the ViT feature extractor, and believe it is a fair and faithful representation of raw perceptral input. There are no additional supervision beyond vision in the DINOv2 pretraining stage, and it therefore, does not encode any semantic regularities (if we agree that such semantic regularities do not emerge automatically during vision-only pretraining).
>
> ### Question 1: Can the model still ‘ground’ shared lexical identity between `<ENV>` and `<LAN>` tokens are removed?
>
> First, we would like to clarify that in our CHILDES experiments, ground and symbol (e.g., book⟨ENV⟩ and book⟨LAN⟩) "are treated as distinct tokens with separate integer indices; that is, the tokenization provides no explicit signal that these tokens are related, so any correspondence between them must be learned during training rather than inherited from their surface form" (Section 3.1). Therefore, there is no shared lexical identity between `<ENV>` and `<LAN>` tokens are removed.
>
>
> ### Question 2: Does grounding transfer to unseen objects or contexts? Specifically, information between token and intrinsicly visualized object.
> This is an interesting question, which we believe that Ma et al. (2023) has already provided in-depth studies. We hypothesize that the mechanism itself, especially the aggregate head, is general. However, it is hard to experiment under our current framework, as the tokenization is predetermined by the training corpus, and it's very hard to add an "unseed" token.
>
> ### References
> [1] Oquab et al. 2024. DINOv2: Learning Robust Visual Features without Supervision. In TMLR.
>
> [2] Ma et al. 2023. World-to-Words: Grounded Open Vocabulary Acquisition through Fast Mapping in Vision-Language Models. In ACL.

---

### Official Review · Reviewer_2rBC · 2025-10-31

**Soundness:** 3
**Presentation:** 3
**Contribution:** 3
**Rating:** 6
**Confidence:** 3

**Summary:**

This paper investigates the emergence of symbol grounding in large language models (LLMs) and vision–language models (VLMs)—the process by which models learn to bind sensory experience to linguistic meaning. Specifically, the authors examine how sensory information in one modality (e.g., visual or environmental) is encoded and represented in the tokenized form of another modality (e.g., textual or linguistic), and how attention heads learn to associate corresponding tokens across modalities. The first part of the paper presents a set of behavioral analyses designed to measure the strength of symbolic grounding. The authors quantify grounding by testing whether a model trained from scratch can predict the appropriate linguistic token given environmental tokens. They further assess how prediction accuracy (or equivalently, surprisal) changes when the matching environmental token is present versus when alternative cues are provided.
The authors also considered potential confounds from simple co-occurrence statistics and conducted extra experiments to show that symbol grounding cannot be fully explained by simple co-occurrence statistics.
The second part of the paper investigates the mechanistic emergence of symbol grounding within attention heads of VLMs and LLMs during training. The authors find that grounding representations tend to concentrate in the middle layers, implemented through an aggregate attention mechanism.
By perturbing specific attention heads, they demonstrate that symbol grounding is causally mediated by these components—perturbations lead to a measurable drop in their proposed grounding metric.

**Strengths:**

* The study is systematic, covering both behavioral and mechanistic analyses, tested on a wide range of LMs.
* The writing is clear and logically structured, making the argument easy to follow.
* The combination of causal interventions with behavioral probing provides convincing evidence for grounding-related mechanisms.

**Weaknesses:**

* Currently, the discussion of how symbol grounding relates to model failure modes (e.g., hallucination) is limited.
* It might also help to connect this work more explicitly to theoretical accounts of symbol grounding in cognitive science and linguistics.

**Questions:**

* It is unclear how attention heads act as a mechanistic pathway for symbol grounding. What is the precise mechanism introduced by these heads? How do earlier-layer heads influence the residual streams that, in turn, affect later layers? Why does grounding typically emerge in intermediate rather than early or late layers?
* Choice of mismatch tokens:  More discussion is needed on how mismatch tokens are selected, especially within the same semantic category, as this could strongly affect the interpretability of the surprisal effects.
* Surprisal distribution:  I am intrigued that the VLMs exhibit non-deterministic surprisal. It would be informative to visualize the distribution of surprisal values across the token dictionary and to analyze how they relate to different semantic categories or levels of abstraction.
* Abstract category representation:  How does grounding behave for hierarchical semantic categories (e.g. elephant -> mammal -> animal)? For example, if the visual scene depicts an elephant but the text says “the animal,” does the attention head for the abstract category (“animal”) emerge earlier or later than the one for the specific concept (“elephant”)?
* Behavioral observation:  it is puzzling that symbol grounding does not emerge in LSTMs. I am curious to hear the author's explanation on this observation.
* What is the relation of this work to the work by Pavlick discussing symbolic grounding in LLMs?

Reference:
_Pavlick E. Symbols and grounding in large language models. Philos Trans A Math Phys Eng Sci. 2023_

---

> ### Author Response · Authors · 2025-11-21
> **Response to Reviewer 2rBC**
>
> We thank the reviewer for their careful and positive assessment of our work. We are encouraged that the study is seen as “systematic, covering both behavioral and mechanistic analyses, tested on a wide range of LMs,” and that the writing is viewed as “clear and logically structured, making the argument easy to follow.” We are grateful that the second part of the paper is interpreted as demonstrating that grounding representations “concentrate in the middle layers, implemented through an aggregate attention mechanism,” and that the “combination of causal interventions with behavioral probing provides convincing evidence for grounding-related mechanisms.”
>
> ### Weakness 1: Discussion on how symbol grounding relates to failure modes is limited
>
> We thank the reviewer for the suggestion.
>
> On how symbol grounding relates to model failure modes, we further analyze how the discovered mechanisms relate to the model's grounding ability (Figure 12: Information Gain vs Aggregate Head Number). Concretely, we examine failure cases where the model fails to ground (i.e., yields low information gain) and find that these are associated with fewer aggregate heads. For each word, we plot information gain against the number of aggregate heads and fit a linear regression. We observe a clear positive correlation: words with more aggregate heads tend to have higher information gain. This relationship is especially strong and close to linear in the CHILDES text-only setting, with ($R^2 = 0.847$).
>
> We completely agree that this is an important direction and consider hallucination detection and mitigation as key directions of our future work. One possible (preliminary) connection can refer to Huang et al. (2024), in which they observe that "a recurring pattern where the inception of many hallucinated contents aligns with the subsequent tokens generated after a columnar attention pattern. Notably, these columnar attention patterns often manifest on tokens that lack substantial informativeness, e.g., full stop or quotation marks" as shown in their Figure 2 and 3. This shows that analysis of attention mechanisms can lead to practical applications. In our case, it would be interesting to detect aggregate heads at scale, and identify if these heads attend to incorrect grounds in cases of hallucination.
>
> ### Weakness 2 and Question 6: Discussion on the relation between this work and cognitive science and linguistics literature, especially Pavlick (2023)
>
> We thank the reviewer for bringing this out, and we are delighted to have this discussion in our updated manuscript (Section 6, marked in blue) as there is another page of space allowed in the rebuttal & revision phase.
> We echo Pavlick (2023) in arguing that *LLMs lack the capacity to represent abstract symbolic structure* should not be accepted a priori. Instead, such claims should be evaluated carefully and empirically, with the focus on uncovering the models’ underlying competence rather than drawing conclusions solely from high-level architectures and surface-level performance.
>
> ### Question 1: How attention heads act as a mechanistic pathway for symbol grounding
>
> Please see Question 1 in our general response for our answers to this question. Related content can be found in Section 5.2 of our paper.

---

> > ### Author Response · Authors · 2025-11-21
> >
> > ### Questions 2--4: Discussions related to semantic categories
> >
> > We thank the reviewer for initiating this discussion. We believe it is worth another paper to investigate these questions systematically, and we are happy to share some preliminary evidence here.
> >
> > - **Question 2: How are mismatched words chosen.**
> > We kindly refer the reviewer to Lines 240-245: we construct the vocabulary with 100 nouns from the MacArthur–Bates Communicative Development Inventories that occur frequently in our corpus. Each word serves once as the target, with the remaining 99 used to construct mismatched conditions. See the complete word list in Appendix A2 and A3.
> >
> > - **Question 3: Visualization of surprisals for different levels of abstraction.**
> > As requested, we have created a visualization of the distribution of surprisal values across the token dictionary for VLM, as shown in Appendix C1 of the updated manuscript (Figure 10).
> > Words in the beeswarm plot are sorted by the grounding information gain in an increasing order. We find from the plot that concrete animals like "duck", "elephant", "cat", "giraffe" are the tokens with the largest information gain (IDs=97-100), while the word "animal" has a smaller information gain (ID=20). The same happens for words like "cheese" (ID=56), "cookie" (ID=67), "cake" (ID=70) and "food" (ID=30).
> >
> > - **Question 4: Attention head w.r.t. semantic hierarchy.**
> > We would like to note that in our context settings, the only way the target token is involved is the prediction of next token. So, if the visual scene is fixed, no matter what we are predicting (animal, cat, elephant), the context are the same, so under your proposal, the attention values, aggregate heads, etc., will not change.
> > That said, we did notice something interesting when comparing across different target tokens. As shown in Appendix C.3, Figure 12, we find that a strong correlation between a target token's grounding information gain and its aggregate head number across all settings we experimented with.
> >
> > ### Question 5: Why didn't symbol grounding emerge in LSTMs?
> > Current results suggest that symbol grounding emerges in Transformers and SSMs but not in LSTMs. We hypothesize that grounding depends on an explicit attention-like mechanism over past states, which allows a model to "look back" at the visual/environmental context and retrieve relevant information there. Transformers and SSMs provide this capability, whereas unidirectional LSTMs face a structural limitation: they must compress the entire history into a single fixed-size hidden state, which creates a bottleneck for effective grounding.

---

### Official Review · Reviewer_7SQR · 2025-11-03

**Soundness:** 2
**Presentation:** 3
**Contribution:** 2
**Rating:** 4
**Confidence:** 4

**Summary:**

The paper is focused on how various neural architectures solve the symbol grounding problem, that is, the problem raised by Hanard 1990 which asks how to symbols in an purely formal symbolic system come to refer to their real-world referents. The authors train several models from scratch on a variety of corpora (including CHILDES, and a visual dialog corpus). The authors manipulate the context provided to the model (either relevant or irrelevant to the utterance to be predicted) and show that during training, Transformers and SSMs see a sudden drop in surprisal in the relevant setting, showing that they develop a mechanism for using the context to reduce prediction uncertainty whereas LSTMs do not. The authors further perform some mechanistic analyses to demonstrate which attention heads are highly involved with this ability.

Overall, I lean against accepting this paper, with the primary reason being the framing. I just don't think the results here really have much to do with symbol grounding (I elaborate in Weaknesses). That said, I do think there are some interesting results--ones I might even share with colleagues--but I think they have more to do with architectures and their ability to handle long-context and binding operations. There is a lot to like about the study (the careful experimental design, the fact that models are trained from scratch, the variety of complementary analyses). But as the paper pitches its primary contribution and novelty as being about grounding per se, I think we are obligated to make sure that this pitch is framed appropriately and accurately. These days, its quite plausible that many readers will learn about the problem the first time in reading this work, and we have an obligation to make sure that we frame and interpret results accordingly.

**Strengths:**

* I appreciate that the authors train models from scratch in a careful way. This permits more conclusive takeaways from the results.
* The variety of datasets chosen is interesting, and adds interesting dimensions beyond standard VQA style experiments
* It is nice to see a combination of types of analyses - behavioral, traning dynamics, mechanistic studies. These combine to give a fuller picture of what we are looking at and form the ingredients of an overall nice paper.
* The comparison across different architectures is interesting. Its a nice clear illustration of the importance of an architectural binding mechanism. I know many people (esp. in cog and neuro science) who would like to see this result.

**Weaknesses:**

I believe the biggest issue with the paper is the framing. I think the results here are not "about" what the authors say the paper is about--i.e., I don't really see these results as being "about" grounding. I view framing in science as all-important, its not mere marketing, it actually dictates what the contribution of the paper is, and what future work it can/should influence. So at a reputable conference like ICLR, there is an obligation to quality control the framing at least as much as the experiments and results, if not more. That is why I recommend rejecting, but with the hope that the authors resubmit with new framing but largely the same results.

To elaborate: The authors study the "grounding" problem by analyzing how well LMs incorporate context to predict the next word in a sequence. The data is always represented as sequences of tokens (the exact source of these tokens varies, but they are always tokens of some kind). The mechanism the authors describe is one in which LMs attend backward to tokens in context and use that to increase/decrease the probability of the next word.

My concern is that this mechanism described is extremely generic, and in fact describes how LMs do most of what they do. The fact that the tokens represent environment vs. language is irrelevant here, and I'd expect the exact same mechanism to apply in other contexts in which the model must retrieve from context....e.g., in a long text passage in which the model needs to produce any co-referential to syntactically related phrases which require looking back in the context. (Actually, I'd recommend that the authors run some controls of this type to convince readers that the mechanism is somehow specific to 'grounding' and not general to 'binding'.)

The above matters even more than it would otherwise because the authors hang heavily on the fact that their work is more faithful to Hanard's definition of symbol grounding. But Hanard's whole point was about how, once you have a system that represents the world as symbols, how do you know you aren't just studying relationships between meaningless symbols. I think his criticism could apply exactly to the work described here (or to most of modern LLMs, its not just this paper). I am pasting the abstract of Hanard 1990 for reference, I think it is pretty apparent why modeling the mechanisms for relating one symbol to another in an LM does not get at what he is going for. In fact, the related work that the authors point to as being merely about correlation/geometry is arguably closer to what Hanard was asking for, rather than something about the (quasi-symbolic) operations carried out by attention heads.

Hanard: "There has been much discussion recently about the scope and limits of purely symbolic models of the mind and about the proper role of connectionism in cognitive modeling. This paper describes the "symbol grounding problem": How can the semantic interpretation of a formal symbol system be made intrinsic to the system, rather than just parasitic on the meanings in our heads? How can the meanings of the meaningless symbol tokens, manipulated solely on the basis of their (arbitrary) shapes, be grounded in anything but other meaningless symbols? The problem is analogous to trying to learn Chinese from a Chinese/Chinese dictionary alone. A candidate solution is sketched: Symbolic representations must be grounded bottom-up in nonsymbolic representations of two kinds: (1) "iconic representations" , which are analogs of the proximal sensory projections of distal objects and events, and (2) "categorical representations" , which are learned and innate feature-detectors that pick out the invariant features of object and event categories from their sensory projections. Elementary symbols are the names of these object and event categories, assigned on the basis of their (nonsymbolic) categorical representations. Higher-order (3) "symbolic representations" , grounded in these elementary symbols, consist of symbol strings describing category membership relations (e.g., "An X is a Y that is Z"). Connectionism is one natural candidate for the mechanism that learns the invariant features underlying categorical representations, thereby connecting names to the proximal projections of the distal objects they stand for. In this way connectionism can be seen as a complementary component in a hybrid nonsymbolic/symbolic model of the mind, rather than a rival to purely symbolic modeling. Such a hybrid model would not have an autonomous symbolic "module," however; the symbolic functions would emerge as an intrinsically "dedicated" symbol system as a consequence of the bottom-up grounding of categories' names in their sensory representations. Symbol manipulation would be governed not just by the arbitrary shapes of the symbol tokens, but by the nonarbitrary shapes of the icons and category invariants in which they are grounded.

tldr: I think the authors have something interesting to say about a generic mechanism via which LMs manage and retrieve relevant information from context to support next word prediction, and how certain architectures (those with a mechanism for content-adressable retrieval) are more capable of this than others. That is a very interesting finding that many would like to read about. But when framed as a paper about grounding, I think it it is misleading and makes higher level claims that are not actually supported by the studies.

**Questions:**

Can you explain why the gather head doesn’t really matter. What is happening instead then? I.e., what is having the actual effect of gathering, in your proposed mechanism, if the gatherer heads aren't causally important?

---

> ### Author Response · Authors · 2025-11-21
> **Response to Reviewer 7SQR**
>
> We sincerely thank the reviewer for the thoughtful and detailed review, and for highlighting several aspects of the work they “might even share with colleagues.” We hope these clarifications and additional experiments address the reviewer's concerns, and we are very open to further suggestions on how to sharpen the scientific positioning of the work. Importantly, there appear to be no technical flaws in the method itself as implied by the reviewer, and the requested updates mainly involve framing and clarifications, which can be incorporated without substantial changes to the paper.
>
> ---
> ### Weakness: the appropriateness of framing this work as grounding
>
> We thank the reviewer for the thoughtful comments. Our explanations and corresponding revisions are provided in the general response **Issue 1** (Why the problems studied in this work faithfully operationalize the definition described by Harnad 1990). In short, we believe there may have been a misunderstanding regarding the purpose of our minimal testbed, the technical implementation and the non-symbolic nature of "visual tokens" in vision-language models, or both. We refer the reviewer to Issue 1 of our general response for a detailed clarification.
>
> In addition, we have also updated the manuscript with causal intervention results on VLMs, supporting that aggregate heads are crucial in implementing grounding across setups.
>
> ---
> ### Question: Why does the gather head not really matter
>
> We would first like to clarify that we did not "propose" a mechanism in this paper. Instead, we assess the hypothesis that an existing mechanism (gather-and-aggregate; Bick et al., 2025) implements symbol grounding, and find only supporting evidence that aggregate heads play a role here.
>
> We kindly refer the reviewer to the general response Question 1.1 ("Why only aggregate heads show significant importance, not gather heads") for a detailed explanation. Here we address the question from several complementary angles:
>
> - **Why can grounding rely only on aggregate heads?** Our working hypothesis is that grounding in our setup is primarily a direct linking operation: information from the ground token is broadcast to a nearby linguistic position (typically the token immediately preceding the grounded linguistic token). Aggregate heads naturally implement this kind of "one-hop" routing: they attend from a small set of receiver tokens to a focused set of source tokens and copy their information forward. In this picture, the effective gathering is implemented by these aggregate heads themselves (and surrounding attention patterns), without requiring an additional, distinct class of gather heads to be causally necessary. As discussed in Bick et al. (2025), gather heads are particularly useful in few-shot in-context learning, where consolidating information from examples is critical; this need not be the dominant mechanism in our grounding testbed.
>
> - **Why does the minimal testbed not need gather heads?** In our minimal CHILDES-style setup, the textual template is semantically light, and the environmental evidence is often sufficient by itself to determine the linguistic form. For example, given the environmental context `<CHI> holds:<ENV> a:<ENV> book:<ENV>` the token `book:<ENV>` alone almost determines the corresponding linguistic token (typically `book:<LAN>`). In such cases, there is little to "gather" from other context tokens; a single aggregate head that directly attends from the linguistic position to the relevant environmental token already solves the task. This explains why we see strong causal importance for aggregate heads but do not observe a comparable effect for gather heads in this minimal regime.

---

### Official Review · Reviewer_P6pG · 2025-11-03

**Soundness:** 3
**Presentation:** 4
**Contribution:** 3
**Rating:** 6
**Confidence:** 3

**Summary:**

This paper investigates the mechanistic basis of symbol grounding in multi-modal language models. The authors introduce a controlled evaluation framework designed to trace how abstract linguistic symbols acquire meaning by being connected to environmental content. The authors construct examples with two distinct representations for each lexical item: "environmental" tokens (ENV) representing its non-verbal (e.g., visual or situational) presence, and "linguistic" tokens (LAN) representing its use in a spoken utterance. The authors measure grounding using "grounding information gain," a surprisal-based metric that quantifies the model's ability to predict LAN tokens when its corresponding ENV tokens are present in the context, compared to when a mismatched ENV tokens are present. The authors show that grounding, as measured by information gain, emerges during training in Transformer and state-space models (Mamba-2) but fails to emerge in unidirectional LSTMs. Using saliency analysis and a "TunedLens" analysis, the paper traces the computations critical for grounding to the middle layers of the network, and then show that these peaks correspond to the location in the models that gather-and-aggregate mechanisms emerge, implicating these mechanisms in symbol grounding. Then using a series of head-level ablations, they reveal that aggregate heads are involved in the grounding effect, while gather heads play a less significant role.

**Strengths:**

1. *Methodology:* The paper uses a minimal, controlled testbed that allows them to distinguish between the "symbol" from its "ground" and allows for more precise, causal analysis.
2. *Confound control:* The authors demonstrate that the emergent grounding is not simply a signature of the models learning shallow co-occurrence statistics. In the analysis presented in Section 4.2, they show that grounding information gain continues to rise as the correlation with co-occurrence statistics ($R^2$) falls.
3. *Strong Mechanistic Account:* The authors successfully build a clear argument for the mechanistic circuit underlying symbol grounding. They first localizes the effect to the middle layers using saliency and probing, then hypothesizes a specific circuit (gather-and-aggregate), and finally confirms the hypothesis with causal interventions..
4. *Generalizability:* The authors demonstrate that their findings are not brittle, and that the grounding effects replicated across different model architectures (Transformers, Mamba-2), model scales (4, 12, 18-layer Transformers), and across different grounding modalities, including text-based dialogue and image-grounded dialogue.

**Weaknesses:**

1. *Hallucination prevention*: Though the authors do mention that such analyses are more difficult to conduct in larger scale VLMs and mention the relevance of this work to hallucination mitigation in VLMs in their discussion, the strength of their contribution would be greatly strengthened if they could demonstrate how interventions on the circuits described here may help to mitigate hallucinations in more naturalistic use cases (e.g., with larger, open-source VLMs).
2. *Gaps in Related Work:* One minor weakness is the papers related work, which misses a highly relevant line of inquiry. The "gather-and-aggregate" mechanism is analogous to mechanisms identified in recent work on symbol processing, variable binding, and abstract reasoning in LLMs. These works identify circuit components analogous to the "gather" and "aggregate" operators discussed in this paper, and situating the work within this literature may help increase the broad appeal of the paper (see Questions).

**Questions:**

1. *Connecting to Symbol Binding Literature:* The related work on the gather-and-aggregate mechanism and symbol grounding overlooks recent work on symbol processing and variable binding in Transformers. Could the authors comment on the relationship between their findings and the mechanisms identified in the following (or similar) papers? These seem highly relevant:
	* `https://arxiv.org/abs/2310.17191`
	* `https://arxiv.org/abs/2406.19501`
	* `https://arxiv.org/pdf/2502.20332`
	* `https://arxiv.org/abs/2409.05448`
2. *Asymmetry in Multimodal Settings*: The paper finds a strong asymmetry in the text-only (CHILDES) domain, where aggregate heads are causally critical but gather heads are not. The authors hypothesize this is because the "input template is semantically light". This makes sense. However, this asymmetry seems less likely to hold in the image-grounded setting in which the "ground" is distributed across many visual tokens, and "gathering" information from relevant patches would seem to be a crucial prerequisite for aggregating it. Do the authors have any preliminary results or speculation on whether this aggregate > gather effect decreases or reverses in the image-grounded dialogue setting?

---

> ### Author Response · Authors · 2025-11-21
> **Response to Reviewer P6pG**
>
> We thank the reviewer for their thoughtful and constructive feedback, and we are encouraged by their positive assessment of our work. We appreciate the recognition that our methodology uses a “minimal, controlled testbed” that separates the symbol from its ground and “allows for more precise, causal analysis.” We are also grateful that the confound-control analysis was seen as demonstrating that emergent grounding “is not simply a signature of the models learning shallow co-occurrence statistics,” with grounding information gain continuing to rise even as its correlation with co-occurrence falls. We are pleased that the reviewers found our mechanistic account “strong.”
>
> ### Weakness 1: How this work connects to hallucination detection and mitigation in industry-scale VLMs
> > [T]heir contribution would be greatly strengthened if they could demonstrate how interventions on the circuits described here may help to mitigate hallucinations in more naturalistic use cases (e.g., with larger, open-source VLMs).
>
> We thank the reviewer for the suggestion. We completely agree that this is an important direction; therefore, we present additional causal analysis (see general response and Section 5.4) on VLMs and consider hallucination detection and mitigation as key directions of our future work.
>
> One possible (preliminary) connection can refer to Huang et al. (2024), in which they observe that "a recurring pattern where the inception of many hallucinated contents aligns with the subsequent tokens generated after a columnar attention pattern. Notably, these columnar attention patterns often manifest on tokens that lack substantial informativeness, e.g., full stop or quotation marks", as shown in their Figure 2 and 3. This shows that analysis of attention mechanisms can lead to practical applications. In our case, it would be interesting to detect aggregate heads at scale, and identify if these heads attend to incorrect grounds in cases of hallucination.
>
> However, as mentioned in Section 6, paragraph 1, we note that a systematic investigation of industry-scale generative VLMs is substantial enough to merit another dedicated research paper or two, instead of justifying a weakness of the present work. Such an analysis is inherently complex due to factors including but not limited to: (1) the register-token effects described by Darcet et al. (2024), and (2) confounding influences introduced by engineering choices outside our control---for example, LLaVA (Liu et al., 2023) uses CLIP's ViT encoder (Radford et al., 2021) for visual feature extraction, and hallucinations may arise within the CLIP ViT itself rather than from the main LLaVA generative architecture.
>
>
> ### Weakness 2 and Question 1: Gaps in related work
>
> We thank the reviewer for bringing out the point of discussing the relation between this work and symbol binding. As mentioned in General Response (Issue 2), we have updated our manuscript (Section 6, marked in blue) to reflect this. Specifically, we added the following discussion:
>
> > While our primary purpose of introducing the minimal testbed (child-direct speech setting; Table 1) is to offer a proxy for understanding grounding, experiments in this setting can be naturally viewed as investigations of symbol binding, the problem that studies how symbols are connected together. This work extends the activation-based study on symbol binding (Feng & Steinhardt, 2024; Dai et al., 2024; Feng et al., 2025) and offers evidence on the attention-head level, showing that aggregate heads are crucial in implementing the mechanism (Table 2). In line with Yang et al. (2025), our work suggests that attention heads are crucial in implementing symbolic structures in LMs, with more controlled and causal evidence.

---

> > ### Author Response · Authors · 2025-11-21
> >
> > ### Question 2: Asymmetry between gather and aggregate heads in multimodal settings
> >
> > Thank you for this thoughtful question. In short, we (i) added causal analyses in the image-grounded setting, which still highlight the importance of aggregate heads, and (ii) found that extending our gather–head definition to image tokens is technically non-trivial.
> >
> > 1. Image-grounded causal interventions. We have run additional causal intervention studies in the image-grounded dialogue setting, analogous to those in the text-only CHILDES experiments. When we intervene on aggregate heads in the image-conditioned layers, model performance degrades substantially, confirming that aggregate heads remain causally important even when the "ground" is distributed across many visual tokens. We will include these results in the revised version.
> >
> > 2. Why we do not report "gather heads" for images. In our original text-only definition, gather heads are characterized relative to discrete ground segments (between a previous and current "gathering" token). In the image case, however, relevant visual tokens for an object are typically contiguous in space, and many tokens in the object region could be treated as valid "ground" tokens. Under this regime, the gather region becomes almost the entire contiguous block, making the notion of "gathering from before the current token and after the previous gather token" degenerate. If we relax the definition and treat "all previous non-gather tokens" as the source for gathering, the bookkeeping requires $O(n^2)$ memory over $n$ patches (up to 0.5T activations in our setting), which makes systematic analysis infeasible within our current infrastructure. In addition, a large portion of visual information aggregation already happens inside the image tokenizer (DINOv2 in our case), so it is unclear how to cleanly separate "gather" vs. "aggregate" behavior at the language-model layers without confounding tokenizer effects.
> >
> > 3. Interpretation of the asymmetry. We therefore restrict our strong aggregate > gather asymmetry claim to the semantically light, text-only CHILDES template, where the definitions are precise and tractable. In line with the reviewer’s intuition, we agree that in fully end-to-end multimodal models one might expect stronger "gather-like" computation over visual patches; a careful characterization of such gather heads in the image-grounded setting is, in our view, an important direction for future work rather than a question we can cleanly resolve in this paper.
> >
> > ### References
> > [1] Darcet et al 2024. Vision Transformers Need Registers. In ICLR.
> >
> > [2] Liu et al. 2023. Visual instruction tuning. In NeurIPS.
> >
> > [3] Radford et al. 2021. Learning transferable visual models from natural language supervision. In ICML.
> >
> > [4] Huang et al. 2023. OPERA: Alleviating Hallucination in Multi-Modal Large Language Models via Over-Trust Penalty and Retrospection-Allocation. In CVPR.
> >
> > [5] Jiahai Feng and Jacob Steinhardt. 2024. How do language models bind entities in context? In ICLR.
> >
> > [6] Qin Dai, Benjamin Heinzerling, and Kentaro Inui. 2024. Representational analysis of binding in language models. In EMNLP.
> >
> > [7] Jiahai Feng, Stuart Russell, and Jacob Steinhardt. 2025. Monitoring latent world states in language models with propositional probes. In ICLR.
> >
> > [8] Yukang Yang, Declan Iain Campbell, Kaixuan Huang, Mengdi Wang, Jonathan D. Cohen, and Taylor Whittington Webb. 2025. Emergent symbolic mechanisms support abstract reasoning in large language models. In ICML.

---

### Author Response · Authors · 2025-11-21
**General Response**

We sincerely thank all reviewers for their insightful and constructive feedback. We first address several common issues raised by multiple reviewers in our general response.

---
### Issue 1: Why the problems studied in this work faithfully operationalize the definition described by Harnad (1990)

*(This issue was raised by Reviewers 7SQR and ZTuF.)*

We would like to clarify that the purpose of the minimal testbed with both environmental and linguistic tokens is to provide a (nearly) fully controlled, noise-free proxy setting that helps understand how grounding is implemented in realistic vision-language models (VLMs). We did not intend to draw decisive conclusions about grounding based solely on results from this minimal testbed without generalized results on VLMs---the settings of the VLMs trained and evaluated in our work were described in Table 1 (final row) of our initial manuscript, which takes both image (nonsymbolic) and text (symbolic) inputs, with the behavioral results shown in Figure 4(b) and (d).

We also realized that the concern from Reviewers 7SQR and ZTuF may come from the fact that generative VLMs (such as the LLaVA architecture used in our work; Liu et al., 2023) represent visual inputs using so-called "visual tokens," and thus may involve some form of symbolic representation. If so, we would like to clarify that unlike symbolic textual token embeddings implemented via lookup tables, these "visual tokens" in VLMs are continuously valued vectors produced by a pretrained neural image encoder, each derived from a square-shaped patch of the input image. Therefore, **a more accurate perspective is that $n$ visual tokens of dimension $d$ constitute an $n \times d$ tensor that represents visual features of an image, rather than $n$ discrete symbolic tokens.** In our work, we use DINOv2 (a vision-only self-supervised image encoder; Oquab et al., 2024) as the visual feature extractor, which outputs high-quality and faithful representation of the raw perceptral input. This design choice avoids a major concern associated with CLIP ViT (the most common practice in VLMs; Radford et al., 2021) that its image representations may inherit symbolic or linguistic regularities introduced during vision-language pretraining.

Additionally, we understand that the results in our initial manuscript may not be fully convincing that VLMs share the same mechanism as models trained on our minimal testbed data, and we are delighted to share the **updated manuscript with full behavioral and mechanistic analyses on the VLM (Section 5.4, marked in blue),** extending experiments on the minimal testbed. All major findings in the minimal testbed hold in the VLM settings. Concretely, our results suggest:

a. **For VLMs, there is also a surge of in saliency flow in the middle layer after a reasonable number of training steps** (Figure 7, right; analogous to Figure 5(a) in the initial manuscript).

b. **Ablating VLM aggregate heads (most found in the middle layers) consistently and significantly increases surprisal compared to controls**, confirming their causal role in linking discrete linguistic symbols to sensorimotor grounds (Figure 7, left; analogous to Table 2 in the initial manuscript).

We believe this update together with the above clarification resolves the concerns of Reviewers ZTuF and 7SQR on the potential gap between our initial manuscript and grounding in Harnad (1990).

---
### Issue 2: Discussion on the connection between this work and symbol binding

*(This issue was primarily raised by Reviewer P6pG and briefly mentioned by Reviewer 7SQR, and more broadly, related to the connection to theoretical work in cognitive science and linguistics raised by Reviewer 2rBC.)*

We thank Reviewer P6pG for mentioning this line of work and for offering the comprehensive reference. We agree that, while our primary purpose of introducing the minimal testbed is to offer a proxy for understanding grounding, the experiments in this setting can be naturally viewed as investigations of symbol binding. We are happy to include discussion on this line of related work (Section 6, marked in blue).

---

> ### Author Response · Authors · 2025-11-21
>
> ### Question 1: How attention heads act as a grounding pathway
>
> *(This line of questions was primarily raised by Reviewers P6pG, 2rBC and zfMv.)*
>
> We would like to clarify that while we drew the hypothesis that the gather-and-aggregate mechanism implements grounding based on behavioral and mechanistic evidence (Section 5.2), our experiment only supports half of it; that is, we only find evidence from causal intervention that supports the importance of aggregate heads (Section 5.3, Results and discussions).
>
> - **Q1.1**: Why only aggregate heads show significant importance, not gather head?
>
>   **A**: Here we offer one possible explanation. We hypothesize that grounding is more of a direct connection process, where information of the ground is directly broadcast to the token immediately preceding the corresponding linguistic token, and aggregate heads naturally implement such a function.
>
>   Additionally, in our minimal testbed, since the context template is semantically light, the environmental evidence alone may suffice to decide the linguistic form. For example, if the environmental context is `<CHI> holds:<ENV> a:<ENV> book:<ENV>`, token `book:<ENV>` is sufficient to determine the linguistic token (which is very likely to be `book:<LAN>`). In such a setting, gathering information from the context is not necessary, as an aggregate head can perform this "one-hop" retrieval directly.
>
> - **Q1.2**: Why does grounding appear on mid layers, instead of shallow or deep layers?
>
>   **A**: We hypothesize that this pattern aligns with established findings on layer-specific functions in Transformer language models (cf. Tenney et al., 2019), where shallow layers primarily capture fine-grained lexical information (e.g., part-of-speech) and deeper layers focus on extracting deep semantics to facilitate next-token prediction. Complementing this view, Liu et al. (2019) report that the middle layers of Transformer models contain the most salient syntactic and general linguistic structure. These prior results and our saliency-based analysis jointly support our interpretation that grounding, which is fundamentally a structure-finding task, is predominantly implemented in the middle layers.
>
> ---
> Please also see our individual response for additional information.
>
> ---
> ### References
> [1] Liu et al. 2023. Visual instruction tuning. In NeurIPS.
>
> [2] Oquab et al. 2024. DINOv2: Learning Robust Visual Features without Supervision. In TMLR.
>
> [3] Radford et al. 2021. Learning transferable visual models from natural language supervision. In ICML.
>
> [4] Tenney et al. 2019 BERT Rediscovers the classical NLP pipeline. In ACL.
>
> [5] Liu et al. 2019 Linguistic Knowledge and Transferability of Contextual Representations. In NAACL-HLT.

---

### Author Response · Authors · 2025-11-27
**Gentle reminder on Paper Discussion**

Dear Reviewers --- we would like to kindly note that there is less than a week to go towards **the end (Tuesday, December 2nd) of the Paper Discussion phase of ICLR**. We would be grateful to have your updated assessments based on our response and revision, and we are happy to answer follow-up questions.

We would also be grateful if AC could help us nudge the reviewers.

---

### Meta-Review · Area_Chair_uVdm · 2025-12-20

**Summary:**

I primarily based my meta-review on the review of Reviewer 7SQR, which I generally agree with. This paper does not seem to have much of anything to do with symbol grounding. It is certainly an interesting paper, I really appreciate the variety of analyses and how many things we are comparing between -architectures, data, etc. But from as early as Figure 1, it's clear that this paper is just trying to analyze a very basic concept in language models, which is how they process information. The reason why I think this is crucial is because the actual techniques being used don't have much merit on their own. Saliency maps have not been a load-bearing component of interpretability/explainability in many years (see https://papers.nips.cc/paper_files/paper/2018/hash/294a8ed24b1ad22ec2e7efea049b8737-Abstract.html), and the causal interventions described in 5.2 do not provide sufficient evidence for the hypothesis because there are too many confounders. Rather, what makes the paper interesting is the connection to symbol grounding. If we removed the text on symbol grounding / binding from the paper and re-evaluate it, I doubt that we would see anything special in this paper. We would be left with a text that proposes some circuits in attention heads in specific models, but then we would have to compare this paper to other circuit papers, and I am afraid that we would find it sorely wanting.

I would encourage the authors to revisit the framing and writing of the paper to see whether this paper can be written without relying on the connection to symbol grounding, which is currently tenuous.

**Reviewer Concerns:**

Reviewer P6pG raised two weaknesses, the second of which is relevant as it pertains to the framing of this work with respect to the broader literature.

Reviewer 7SQR raised the main weakness, which is still outstanding.

Reviewer ZTuF also has the same objection, that the paper doesn't engage with symbol grounding, which is still outstanding.

Reviewer zfMv is not convinced by the mechanistic interpretation, which is still outstanding.

**Reviewer Scores:**

It is very hard for me to forecast how the reviewers would have changed their scores. In general I doubt that the reviewers would have changed their scores because I found the authors' rebuttal unconvincing.

---

### Decision · Program_Chairs · 2026-01-26

Reject